computational chemistry/biochemistry

TDP-43 protein, RRM1 and RRM2 domains, molecular dynamics simulation, aggregation

**Authors for correspondence:**
Chaoqun Li
e-mail: lichaoqun@hdc.edu.cn
Guangju Chen
e-mail: gjchen@bnu.edu.cn

This article has been edited by the Royal Society of Chemistry, including the commissioning, peer review process and editorial aspects up to the point of acceptance.

# Insights into the aggregation mechanism of RNA recognition motif domains in TDP-43: a theoretical exploration

Wei Liu[1], Chaoqun Li[2], Jiankai Shan[1], Yan Wang[1] and Guangju Chen[1]

[1]Key Laboratory of Theoretical and Computational Photochemistry, Ministry of Education, College of Chemistry, Beijing Normal University, Beijing 100875, People's Republic of China
[2]Hebei Key Laboratory of Heterocyclic Compounds, College of Chemistry, Chemical Engineering and Materials, Handan University, Handan 056005, Hebei Province, People's Republic of China

CL, 0000-0002-5888-4182; GC, 0000-0002-1921-9390

The transactive response DNA-binding protein 43 (TDP-43) is associated with several diseases such as amyotrophic lateral sclerosis (ALS) and frontotemporal lobar degeneration (FTLD) due to pathogenic aggregations. In this work, we examined the dimer, tetramer and hexamer models built from the RRM domains of TDP-43 using molecular dynamics simulations in combination with the protein–protein docking. Our results showed that the formations of the dimer models are mainly achieved by the interactions of the RRM1 domains. The parallel β-sheet layers between the RRM1 domains provide most of the binding sites in these oligomer models, and thus play an important role in the aggregation process. The approaching of the parallel β-sheet layers from small oligomer models gradually expand to large ones through the allosteric communication between the α1/α2 helices of the RRM1 domains, which maintains the binding affinities and interactions in the larger oligomer models. Using the repeatable-superimposing method based on the tetramer models, we proposed a new aggregation mechanism of RRM domains in TDP-43, which could well characterize the formation of the large aggregation models with the repeated, helical and rope-like structures. These new insights help to understand the amyloid-like aggregation phenomena of TDP-43 protein in ALS and FTLD diseases.

# 1. Introduction

Many human neurodegenerative diseases are correlated with the abnormal accumulation of proteins in the neurons tissues, such as amyotrophic lateral sclerosis (ALS), Parkinson's disease (PD), Alzheimer's disease (AD) and frontotemporal lobar degeneration (FTLD) etc. [1–3]. Recent evidence has further connected the progressive disorder diseases of ALS, FTLD and AD to the aggregation or polymer formation of proteins [4–7]. The transactive response RNA-binding protein 43 (TDP-43) is not only a primarily nuclear RNA-binding protein but also one of the potential aggregation proteins [8]. It has been evidenced that aggregation in nervous tissues of wild-type TDP-43 is a potential risk in ALS and FTLD [9–11]. Therefore, investigation of TDP-43 aggregation mechanism has gained momentum in recent years [12].

TDP-43 was first discovered in 1995 and is implicated in mRNA metabolism, such as mRNA transport, mRNA stability, pre-mRNA splicing and miRNA processing [8,13]. TDP-43 contains 414 amino acids and consists of one N-terminal domain (NTD), two repeated RNA recognition motif domains (RRM domains), one C-terminal domain (CTD) and linking loops [14]. The domain schematic is shown in figure 1a. Specifically, NTD (residues 1–101) includes one α-helix, six β-strands and several loops, while CTD (residues 274–414) contains a helix-turn-helix structure and unstructured loops [15,16]. RRM domains feature two independent domains of RRM1 (residues 102–176) and RRM2 (residues 192–267) with each including two α-helixes (α1, α2), five β-strands (β1–β5) and a α-linker loop between them (figure 1b) [17,18]. The aggregation phenomena mostly mediated by independent NTD and CTD have been extensively investigated *in vivo* and *in vitro* [19–21]. Some investigated evidence showed that the CTD, included in the fragments of ALS-associated TDP-43 inclusions, is classified as prion-like, and is aggregation prone both *in vitro* and in cell [22,23]. Other studies predicted that the NTD was predicted to adopt a stable fold and was found to drive the formation of large oligomers [24,25]. For instance, Miguel Mompeán and Douglas V. Laurents' group provided specific structural evidence of CTD in TDP-43 with β-hairpins assembling into a parallel β-turn for amyloid-like structure, via electron microscopy (EM), X-ray diffraction and solid-state NMR techniques [26]. Nicolas L. Fawzi's group elucidated the structural basis of dynamic granule formation of TDP-43 via its CTD through ALS-associated mutations [27]. Then, the NMR structure of head-to-tail NTD dimer was solved by them, which induces the native polymerization [28]. Using the NMR and other analytical methods, the studies of Emanuele Buratti and Douglas V. Laurents' group strongly supported that a stably folded NTD is essential for correct TDP-43 function and also enhances dimerization, which is in relation to the pathological aggregation and protein's activities [15].

Recently, some studies have shown that the RRM domains will not only help the interaction between TDP-43 and nucleic acids, but also tend to self-assembling using the nucleic acid binding sites as nucleic acids are absent, which also plays an important role in pathological aggregation of TDP-43 [29–32]. Especially, Elsa Zacco and Annalisa Pastore's group investigated the supramolecular conformations constructed by RRM1 and RRM1-2 through the CD analysis, the ThT-binding assay and the AFM techniques. By analysing the resulting formations after the protein constructing to fully aggregate after 72 h, they found that the aggregates formed by RRM1 or RRM1-2 appeared uniformly thick or softer, almost cotton-like with an average thickness of 365 or 173 nm, respectively [33]. In 2019, they validated that the stabilization of the structure of the two RRM domains and the aggregation propensity are simultaneously increased through the D169G mutation adjacent to RRM1, which also reduces the RNA binding tendency. The mutation causes an insistent exposure of the RNA-binding interface, which not only enhances the aggregation potential but also promotes the production of toxic aggregates [34]. Moreover, the studies of Han-Jou Chen and Christopher E. Shaw's group in 2019 showed that the TDP-43 mutations, such as the RRM adjacent K181E mutation, will largely improve aggregation and are possibly served as pathogenic since they promote wild-type TDP-43 to aggregate dominant-negatively [35]. Jianxing Song's group revealed that the TDP-43 RRM12 domains without ATP became opaque during the incubation, which indicated formation of fibrils via the analysis of heteronuclear single quantum coherence (HSQC) peaks and the fluorescence characteristics [36]. It has been found by David Eisenberg's group that amyloid-like fibril formation presents a runaway domain swap through exchanging the C-terminal β-strands between two monomers from the sample of RNase A with $Q_{10}$ expansion [37,38]. In addition, RNA was shown to be an important factor in preventing aggregation of TDP-43 resulting from the fact that RRM domains with RNA binding will maintain TDP-43 solubility [34–36].

Molecular dynamics (MD) simulations of two mutated structures (D169G and K263E) of RRM domains have been reported, which predicted the mutation leading aggregation tendency [39]. Protein–protein docking studies indicated that the cysteine residues in NTD contribute to aggregation

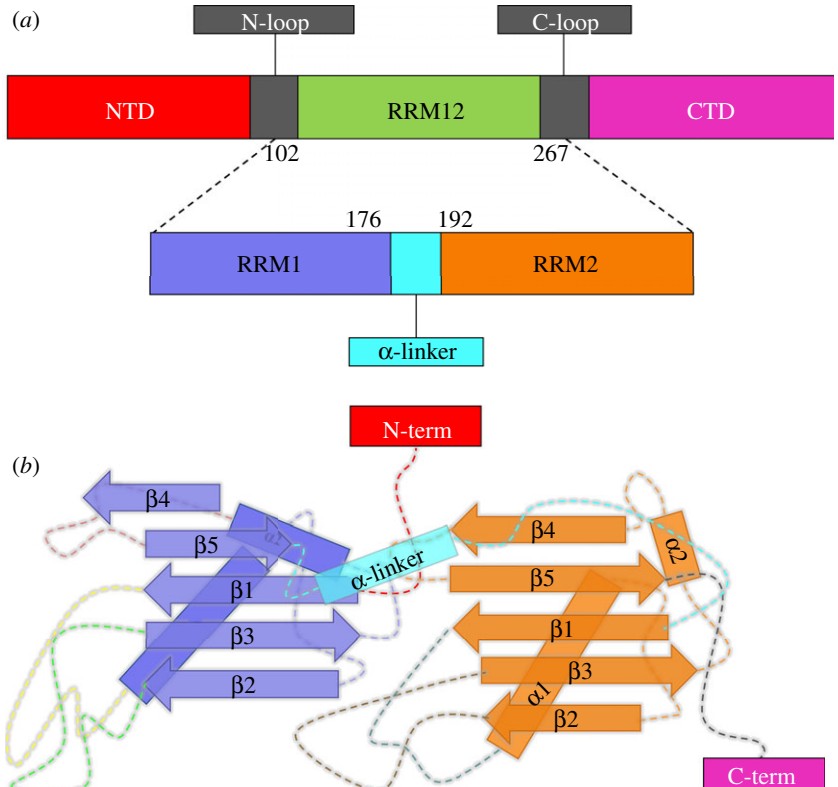

**Figure 1.** Compositions of TDP-43 protein. (*a*) The RRM12 domains are connected to NTD (in red) and CTD (in magenta) by the N-loop and C-loop, respectively. RRM12 domains contain the RRM1 (in purple) domain and RRM2 (in orange) domain that are connected by a α-linker (in cyan). (*b*) The domain organization of the RRM12 domains are represented by the five β-strands (β1, β2, β3, β4, β5) and two mutually vertical α-helixes (α1, α2) of each RRM1 (in purple) domain and RRM2 (in orange) domain, respectively.

propensity [40]. Atomic MD simulation studies elucidated the structural unfolding and stability of NTD and RRMs, which indicated that these unfolded intermediate structures induce the aggregation of TDP-43 [41,42]. MD simulation studies have also shown that the 341–366 segment of CTD was able to drive aggregation of the entire TDP-43 protein [43]. Feng Ding and co-workers theoretically studied the oligomerization of full length hIAPP through multiple molecular systems of increasing number of peptides and suggested that oligomers larger than a trimer allowed the formation of more stable β-sheets of the human islet amyloid polypeptide [44]. However, theoretical studies on the aggregation structures of RRM domains of wild TDP-43 are scarce so far. Especially, the detailed aggregation processes by which small oligomers evolve into large oligomers leading to an amyloid-like structure of TDP-43 are elusive but necessary to understand the aggregation mechanisms of RRM1 and RRM2 domains. To gain an in-depth understanding of the structural and dynamical basis for the aggregation mechanism and the oligomerization process of RRM domains in TDP-43, we used protein–protein docking technique to build the initial structures of dimer, tetramer and oligomer models, and carried out MD simulations with free energy calculations at the atomic levels for these models. Our main objectives were: (i) to establish as many small oligomer models as possible of RRM1 and RRM2 domains, calculate and compare their stabilities; (ii) to elucidate the possible aggregation mechanism by which small oligomers promote large oligomers. This study will help better understanding the RRM domains amyloid-like aggregation mechanism.

# 2. Methods

## 2.1. Model construction for the RRM domains of TDP-43

Based on the previous experimental studies, two tandem RRM1 and RRM2 domains of TDP-43 protein play an important role in the TDP-43 protein aggregation [33–35]. So, the two RRM1 and RRM2 domains were chosen to investigate the aggregation mechanism of this protein. The initial structure of the tandem RRM1

and RRM2 domains was constructed by using the nuclear magnetic resonance (NMR) structure of the RRM domains of TDP-43 protein with RNA binding (Protein Data Bank (PDB) code: 4BS2) (assigned as RRM12 model). In order to eliminate the disturbance on the intra-domain structure from RNA binding, the coordinates of the RRM1 and RRM2 domains were corrected by the NMR structures of the two isolate RRM1 and RRM2 domains (PDB codes: 2CQG and 1WF0). Namely, first, the basis structure of RRM12 model was taken from the NMR structure of the RNA binding RRM domains (PDB code: 4BS2) with the direct superposing of isolate RRM1 and RRM2 on the corresponding positions; second, the corrected coordinates of RRM1 and RRM2 domains captured from that of the NMR structure of isolate RRM1 and RRM2 (PDB code: 2CQG and 1WF0) were merged into the corresponding positions of NMR structure of the RNA binding RRM domains (PDB code: 4BS2); finally, besides removing the bound RNA structure, the full residue coordinates from Lys102 to Asn267 of the RRM1 and RRM2 domains and the linking loop of Asn179 to Val193 between the two domains in the RRM12 model were constructed by the NMR experimental data (PDB code: 4BS2) with the deletion of the extra residues in the RRM1 domain and with the mutated Gly200 residue recovered by the original Glu200 residue in the RRM2 domain for recovering the primary sequence of this domain [45]. Five initial dimer models were produced from the average structure of the monomer RRM12 model by using the protein–protein docking software [46–48]. The tetramer models were constructed based on the average structure of the most stable dimer model using the same method. The large aggregation models, such as hexamer, octamer, dodecamer and so on, were constructed using the superimposing-creating method. The details of construction processes for these models can be found in §§3.2, 3.3.1 and 3.3.2, respectively. For MD simulation, each model system was explicitly solvated by using the transferable intermolecular potential 3P (TIP3P) water inside a rectangular box large enough to ensure the solvent shell with the depth of 14 Å in all the six directions of the model.

## 2.2. Molecular dynamics simulation and ZDOCK

MD simulations for these models, including minimization, heating and equilibration, were performed by using the sander module of AMBER16 package with a classical AMBER parm99 [49,50]. The details of the MD procedure are given in the electronic supplementary material. The Cpptraj and Ptraj utilities of the AMBER16 program were used to obtain root-mean-square deviation (RMSD), average structures, hydrogen bond and hydrophobic interactions etc. [51]. ZDOCK algorithm based on fast Fourier transform searches all possible binding modes by changing the translational and rotational positions of proteins; and each mode was evaluated by a comprehensive energy scoring function that includes some terms of shape complementarity, electrostatics and statistical potential [52–54]. ZDOCK software was used to search the possible binding conformations of these models (v. 3.0.2, http://zdock.umassmed.edu/) [46,52–54]. The details of the docking procedure are available in the electronic supplementary material.

## 2.3. Binding free energy analysis

The molecular mechanics Poisson-Boltzmann surface area (MM-PBSA) method was used to perform the binding free energy analysis in AMBER16 package [55,56]. The binding free energy ($\Delta G_{\text{binding}}$) was calculated based on the followed equation:

$$\Delta G_{\text{binding}} = G_{\text{complex}} - (G_{\text{ligand}} + G_{\text{receptors}}), \tag{2.1}$$

where $G_{\text{complex}}$, $G_{\text{ligand}}$ and $G_{\text{receptors}}$ are the free energies of complex, ligand and receptor calculated from the snapshots of the MD trajectories, respectively. The binding free energy was calculated as follows:

$$\begin{aligned} \Delta G_{\text{binding}} &= \Delta H_{\text{binding}} - T\Delta S \\ &= \Delta E_{\text{MM}} + \Delta G_{\text{solv}} - T\Delta S. \end{aligned} \tag{2.2}$$

The details of the MM-PBSA method can be found in the electronic supplementary material.

# 3. Results

## 3.1. Structural characteristics of the monomer RRM12 model

For determining the structure of the RRM1 and RRM2 domains, the 1000 ns MD simulation for the RRM12 model was carried out. The root-mean-square deviation (RMSD) values of all backbone atoms

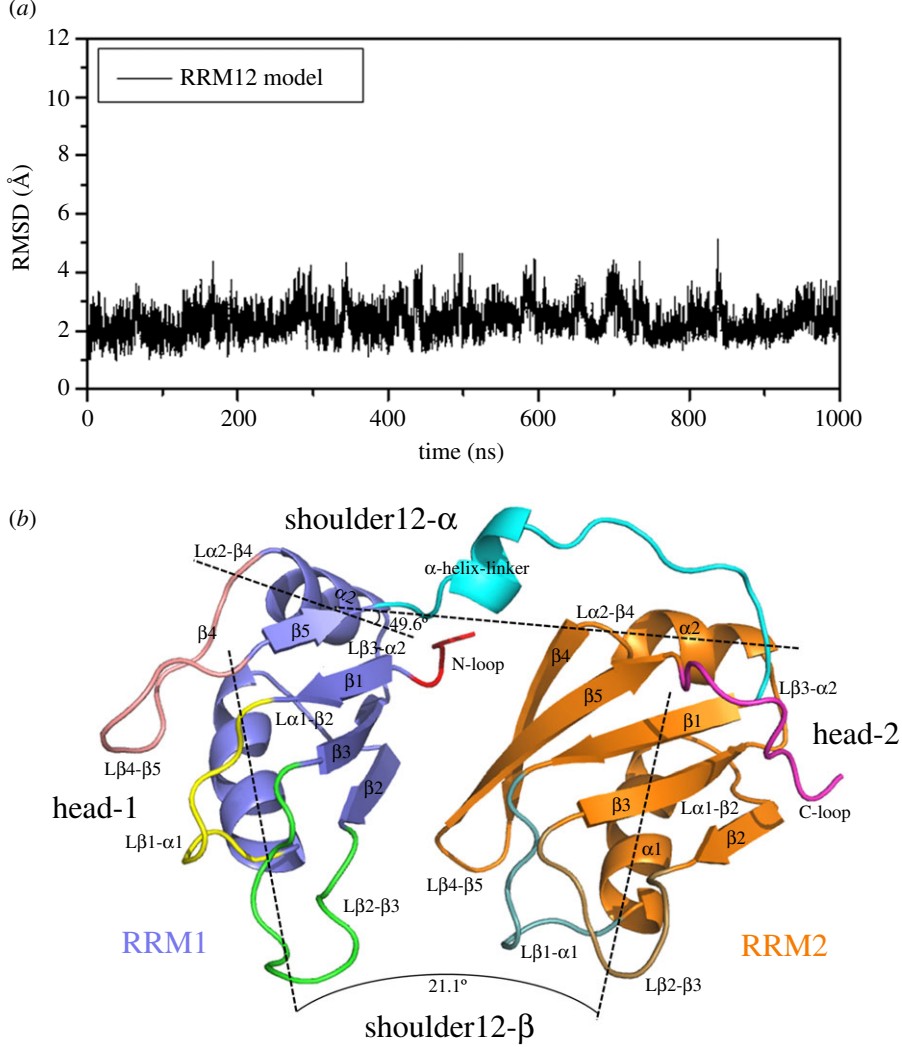

**Figure 2.** (*a*) RMSD values of heavy atoms with respect to the initial structure of the RRM12 system. (*b*) The average structure of the RRM12 system. The RRM1 domain and the RRM2 domain are coloured in purple and orange, respectively.

of the RRM1 and RRM2 domains referenced to the corresponding starting structure over the trajectory of RRM12 model were examined to determine if the system had attained equilibrium, and are shown in figure 2*a*. It can be seen from figure 2*a* that the RRM12 model reached equilibrium after short simulation times, and the energies were found to be stable during the remainder of the simulation. Therefore, the equilibrated conformation for this model was extracted from the trajectory analysis. The corresponding average structure of RRM12 model is shown in figure 2*b*. In this average structure, the RRM1 and RRM2 domains are linked by a small α-helix as the flexible linking loop. Both domains display the similar conformation, i.e. the five β-strands of β1, β2, β3, β4, β5 form a curve sheet layer nearby the plane constructed by two almost vertical α1-α2 helixes; the angles of the α1-α2 helixes and the mass centre distances between the α1–α2 helixes plane and the β-strands sheet layer are 104.3° and 98.2°, 8.2 and 7.6 Å in RRM1 and RRM2 domains, respectively. Each β-strand and each α-helix in the two domains are linked by the corresponding linking loop, named by the two linked units as Lβ1-α1, Lα1-β2, Lβ2-β3, Lβ3-α2, Lα2-β4 and Lβ2-β5. The linking order is following by β1-α1-β2-β3-α2-β4-β5 in each domain with the β4 strand representing a loop structure in the RRM1 domain (figure 2*b*). The structural characteristics of each domain in the RRM12 model show a 'hotdog' fold sketch, such that the curve β-sheet wraps up two α-helices in one side. The relative position of two domains can be described in detail, by the mass centre distance, and the relative rotation angles of the two α1 helices and the two α2 helices in RRM1 and RRM2 domains, to be 24.9 Å, 21.1° and 49.6°, respectively. It can be seen from the results that the two domains linked by the small α-helix linking loop lay nearby each other with the certain spiral and tilt angle (figure 2*b*). Especially, comparing the above calculated distance and angles to the experimental data of 23.9 Å, 16.4° and 24.2° in the RNA binding RRM

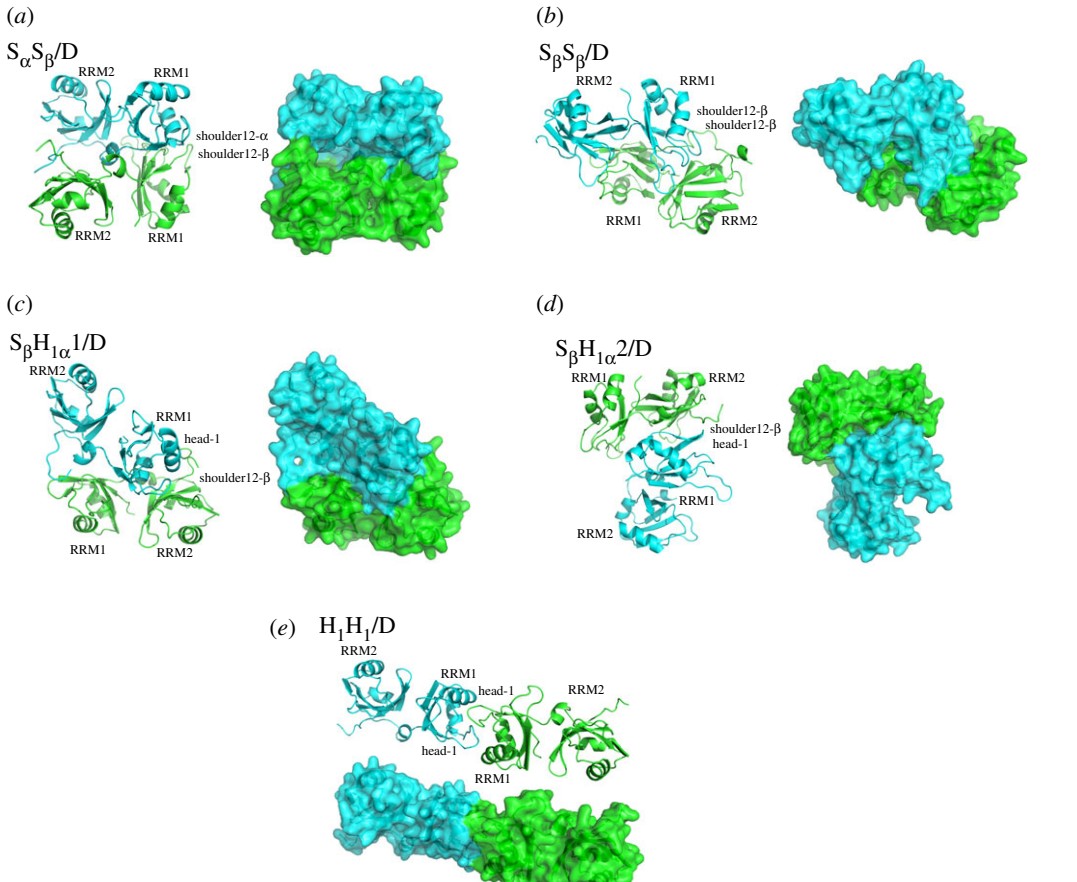

**Figure 3.** Average structures and the corresponding non-transparent surfaces of (*a*) dimer $S_\alpha S_\beta$/D, (*b*) $S_\beta S_\beta$/D, (*c*) $S_\beta H_{1\alpha}1$/D, (*d*) $S_\beta H_{1\alpha}2$/D and (*e*) $H_1 H_1$/D systems. Two monomers in the five dimer models are coloured in cyan and green.

domain structure, respectively, it is found that the RNA binding causes the decrease of spiral angle amount between the two α2 in two domains and results in the tendency of two domains paralleling with each other. Also, it can be seen that the current RRM12 model constructed by the superimposing technique reproduces the experimental structures [18]. This calculated monomer RRM12 model will be prepared to construct the oligomer aggregation models for the investigation of the aggregation mechanism of RRM domains of TDP-43 protein.

## 3.2. The structure and interaction characteristics of RRM domain dimer models

To explain the aggregation process of RRM domains of TDP-43, the initial dimer models were produced from the average conformation of the calculated RRM12 model by protein–protein docking method using ZDOCK software (http://zdock.umassmed.edu/) [46]. Based on this method, five dimer models were screened from the top 250 docking results with the high ZDOCK scores and the groups of some similar contacted modes. The details of the docking procedure are available in the electronic supplementary material. Finally, MD runs of 1000 ns for five models were carried out by following the same protocol above.

### 3.2.1. Structure analysis of the dimer models

Based on the similar trajectory analysis, MD simulations of 1000 ns for all five dimer models reached equilibrium in the time range of 400–600 ns, and the corresponding energies were found to be stable during the remainder of the simulation. Therefore, the equilibrated conformations have been extracted from 800 to 1000 ns of simulation time, and are shown in figure 3 with the corresponding non-transparent surface structures. The RMSD values of all backbone atoms referenced to the corresponding starting structures over the trajectories of the five dimer models are shown in electronic supplementary material, figure S1a–e. Based on the asymmetrical ellipsoid shape of the monomer RRM12 model

(figure 2*b*), four outer edges of each model were defined as 'shoulder12-α', 'shoulder12-β', and 'head-1', 'head-2', which will be applied to easily identify the types and names of the five dimer models. The shoulder12-α in each model represents the outer edge consisted of the α2-helix in RRM2 domain, and the linking loop (or the α-helix-linker) between RRM1 and RRM2 domains; the shoulder12-β represents the outer edge consisted of two β-sheet layers in RRM1 and RRM2 domains; then head-1 represents the outer edge constructed by the α1, α2 helices of the RRM1 domain, and head-2 represents the outer edge constructed by the β-sheet layer tail of the RRM2 domain (figure 2*b*). From the above, the five dimer models were defined as $S_\alpha S_\beta/D$, $S_\beta S_\beta/D$, $S_\beta H_{1\alpha}1/D$, $S_\beta H_{1\alpha}2/D$ and $H_1 H_1/D$ models with three classified groups of shoulder-to-shoulder, i.e. $S_\alpha S_\beta/D$ and $S_\beta S_\beta/D$ models, shoulder-to-head, i.e. $S_\beta H_{1\alpha}1/D$ and $S_\beta H_{1\alpha}2/D$ models, and head-to-head, i.e. $H_1 H_1/D$ model, respectively (figure 3).

It can be seen from the average structure in figure 3*a* that the $S_\alpha S_\beta/D$ model presents that the α-helix linking loop and β5 strand of RRM1 domain in the shoulder12-α side of one monomer RRM12 model (cyan in figure 3*a*) combines face-to-face with the curve β-sheet layer of RRM1 domain in the shoulder12-β side of the other monomer RRM12 model (green in figure 3*a*). Meanwhile, the α-helix linking loop of one monomer occupies the interstice of two domains of the other monomer, and vice versa. In figure 3*b*, the $S_\beta S_\beta/D$ model presents that the curve β-sheet layer of the RRM1 domain in the shoulder12-β side of one monomer RRM12 model (cyan in figure 3*b*) combines interlaced with the β-sheet layer of RRM1 in the shoulder12-β side of the other monomer RRM12 model (green in figure 3*b*). The above two dimer models were formed by the interaction between the shoulder sides, and thus were called the group of shoulder-to-shoulder. The $S_\beta H_{1\alpha}1/D$ model in figure 3*c* presents that the α2-helix of RRM1 domain in the head-1 side in one monomer RRM12 model (cyan in figure 3*c*) inserts tilted into the interstice of two β-sheet layers, nearby the β-sheet layer of RRM1 domain, in the shoulder12-β side of the other monomer RRM12 model (green in figure 3*c*). Moreover, the linking loop in the same monomer RRM12 model interacts with the β-sheet layer of RRM1 in the shoulder12-β side of the other monomer RRM12 model. Then, the $S_\beta H_{1\alpha}2/D$ model in figure 3*d* presents that the α2-helix of RRM1 domain in the head-1 side in one monomer RRM12 model (cyan in figure 3*d*) inserts vertically into the interstice of two β-sheet layers in the shoulder12-β side of the other monomer RRM12 model (green in figure 3*d*). So, the $S_\beta H_{1\alpha}1/D$ and $S_\beta H_{1\alpha}2/D$ dimer models were called the group of shoulder-to-head. The $H_1 H_1/D$ model in figure 3*e* presents that the α1-helix of RRM1 domain in the head-1 side in one monomer combines with the loose β4 and β5 in the other monomer, and vice versa, as the group of head-to-head. In conclusion, the combinations of two RRM1 domains dominate the formations of these dimer models, which is consistent with the experimental results of RRM1 function in the TDP-43 protein aggregation [33].

### 3.2.2. Analyses of binding free energies and interactions of the dimer models

To address the combination abilities for these five dimer models of $S_\alpha S_\beta/D$, $S_\beta S_\beta/D$, $S_\beta H_{1\alpha}1/D$, $S_\beta H_{1\alpha}2/D$ and $H_1 H_1/D$, their binding free energies have been carried out by using the MM-PBSA method based on their corresponding MD simulations. All energy terms and the total binding free energies for these models are given in table 1. It can be seen that the binding free energies for $S_\alpha S_\beta/D$, $S_\beta S_\beta/D$, $S_\beta H_{1\alpha}1/D$, $S_\beta H_{1\alpha}2/D$ and $H_1 H_1/D$ dimer models are −57.66, −41.97, −23.11, −17.80 and −4.22 kcal mol$^{-1}$, respectively, which reasonably show the stability and possible existence of the five dimer models from the favourable binding energies. This result supports the experimental suggestion that the RRM domains can possibly cause the aggregation of TDP-43 proteins themselves under the absence of RNA [33,34,57]. Moreover, based on the differences of the energies of the five dimer models, their combination abilities between two monomer RRM12 models along the shoulder sides are obviously greater than that along the head sides, which proposes that the further aggregation along the shoulder sides would be favourable.

To further explore the interaction mechanisms between the two RRM12 monomer models in these dimers with the different binding ways, the percentages of occurrences of all possible hydrogen bond and hydrophobic interactions located at the interfaces for the five dimer models were extracted from the MD simulations, and are shown in table 2. The criteria include a donor–acceptor distance of <3.5 Å and a donor–proton–acceptor angle of greater than 120° for a hydrogen bond, while a C-C distance of less than 4.5 Å for a hydrophobic interaction. It can be seen from table 2 that the total occupancies of hydrogen bond and hydrophobic interaction are 628.64% and 554.72% for $S_\alpha S_\beta/D$ model, 302.29% and 638.81% for $S_\beta S_\beta/D$, 520.81% and 431.48% for $S_\beta H_{1\alpha}1/D$, 190.74% and 593.42% for $S_\beta H_{1\alpha}2/D$, and 215.96% and 112.83% for $H_1 H_1/D$, respectively. Obviously, the total occupancies for the hydrogen bonds or hydrophobic interactions for $S_\alpha S_\beta/D$, $S_\beta S_\beta/D$, $S_\beta H_{1\alpha}1/D$ and $S_\beta H_{1\alpha}2/D$

**Table 1.** Components of the MM-PBSA free energies (kcal mol$^{-1}$) for the $S_\alpha S_\beta$/D, $S_\beta S_\beta$/D, $S_\beta H_{1\alpha}$1/D, $S_\beta H_{1\alpha}$2/D and $H_1H_1$/D dimer models. $\Delta G_{np} = \Delta E_{vdw} + \Delta G_{np/solv}$, $\Delta G_{pb} = \Delta E_{ele} + \Delta G_{pb/solv}$, $\Delta H_{binding} = \Delta G_{np} + \Delta G_{pb} + \Delta E_{int}$, $\Delta G_{binding} = \Delta H - T\Delta S$.

| model | $S_\alpha S_\beta$/D | $S_\beta S_\beta$/D | $S_\beta H_{1\alpha}$1/D | $S_\beta H_{1\alpha}$2/D | $H_1H_1$/D |
|---|---|---|---|---|---|
| $\Delta E_{ele}$ | −574.68 | −160.57 | −307.64 | −252.14 | −188.48 |
| $\Delta E_{vdw}$ | −147.57 | −146.63 | −155.24 | −74.73 | −58.48 |
| $\Delta E_{int}$ | 0.00 | 0.00 | 0.00 | 0.00 | 0.00 |
| $\Delta G_{np/solv}$ | −19.14 | −18.15 | −19.26 | −10.29 | −7.05 |
| $\Delta G_{pb/solv}$ | 633.36 | 231.30 | 397.46 | 279.29 | 209.91 |
| $\Delta G_{np}$ | −166.71 | −164.78 | 378.20 | 269.00 | 202.86 |
| $\Delta G_{pb}$ | 58.69 | 70.73 | −174.50 | −85.02 | −65.53 |
| $\Delta H_{binding}$ | −108.02 | −94.05 | −79.86 | −57.87 | −44.10 |
| $T\Delta S$ | −50.36 | −52.08 | −61.57 | −33.23 | −39.88 |
| $\Delta G_{binding}$ | −57.66 | −41.97 | −23.11 | −17.80 | −4.22 |

models along the shoulder sides in RRM1 domains between two monomer RRM12 models are larger than those for $H_1H_1$/D one along the head sides, which are consistent with the binding free energy calculations. Some binding sites have been found at the corresponding interfaces with the binding type of shoulder or head for these models. For example, the interaction residues of hydrogen bonds and hydrophobic interactions for $S_\alpha S_\beta$/D model locate at Ser163, Asp174, Cys175, Lys176, Gln182, Gln184, Asp185, Arg227 and Met162, Leu177, Pro262 in the shoulder12-$\alpha$ side of RRM1 domain of one RRM12 monomer (cyan in figure 3a), and Lys145, Arg165, Lys176, Asn179, Gln182, Arg227, Glu261 and Leu109, Phe147, Phe149, Pro225 in the shoulder12-$\beta$ side of RRM1 domain of the other RRM12 monomer (green in figure 3a), respectively. Moreover, the ionic interaction analysis for $S_\alpha S_\beta$/D, $S_\beta S_\beta$/D, $S_\beta H_{1\alpha}$1/D, $S_\beta H_{1\alpha}$2/D and $H_1H_1$/D models were calculated, and are shown in figure 4. Similarly, the criteria for an ionic interaction include a distance of less than 5.0 Å between the HN group of positive Arg or Lys and the O atom of negative Asp or Glu. It can be seen that the ionic interactions mainly occurred at the residue-pairs of Asp174-Lys176 and Glu156-Arg171 with the total occupancy of 161.1% for $S_\alpha S_\beta$/D model, the residue-pairs of Glu186-Arg171 and Lys140-Glu200 with the total occupancy of 104.1% for $S_\beta S_\beta$/D model, the residue-pairs of Asp185-Lys136 and Asp174-Arg165 with the total occupancy of 175.9% for $S_\beta H_{1\alpha}$1/D model, the Glu156-Arg171 residue-pair with the occupancy of 51.6% for $S_\beta H_{1\alpha}$2/D model. Conversely, there is no ionic interaction for $H_1H_1$/D model. These results are consistent with the analyses of binding energies and interactions discussed above. The non-transparent surfaces in figure 3 show the binding interfaces in the studied dimer models, and as expected the rank of the binding areas in this interface is in consistent with the order of the binding free energies and interactions in this interface. In conclusion, the greater the combination ability, the larger the binding area.

To compare the differences of the energies and interactions of the dimerization, and to further provide the quantitative information of energy contributions of the residues, the decompositions of the corresponding binding free energies into the key residues were performed for the five dimer models, and are shown in figure 5a and b only for $S_\alpha S_\beta$/D and $H_1H_1$/D dimer models, respectively. It can be seen from the energy decompositions of two models with the energy contributions more than 2 kcal mol$^{-1}$ (i.e. the decomposition energy values less than −2 kcal mol$^{-1}$) that the distinct differences of energy decompositions occur mainly at the crucial residues of Glu156, Met162, Ser163, Asp174, Leu177, Asp185, Arg227, Pro262 in one monomer (i.e. the cyan one in figure 3a) and Leu109, Phe147, Phe149, Arg165, Arg171, Asn179, Gln182, Phe221, Phe231 in the other one (i.e. the green one in figure 3a) for $S_\alpha S_\beta$/D dimer model, and at the crucial residues of Lys114, Met167, Ile168 and Lys114, Arg165, Met167, Trp172 for $H_1H_1$/D dimer model (figure 3e). Obviously, the crucial interaction sites in $S_\alpha S_\beta$/D dimer model are more than those in $H_1H_1$/D one. The binding residue numbers and locations support the binding free energy calculations, and the results of hydrogen bonds and hydrophobic interactions. Other details of the decompositions for other dimer models can be found in electronic supplementary material, figure S2.

**Table 2.** Occupancies (%) of hydrogen bonds and hydrophobic interactions between different RRM12 monomers for the $S_\alpha S_\beta/D$, $S_\beta S_\beta/D$, $S_\beta H_{1\alpha}1/D$, $S_\beta H_{1\alpha}2/D$, $S_\beta H_{1\alpha}2/D$ and $H_1 H_1/D$ models.

| $S_\alpha S_\beta$/D | | occupancy% | | $S_\beta H_{1\alpha}2$/D | | occupancy% | |
|---|---|---|---|---|---|---|---|
| **hydrogen bonds** | | | | **hydrogen bonds** | | | |
| Lys 176 | 182 Gln | NZ-HZ* … OE1 | 84.25 | Ser 163 | 197 Arg | O … H*-N* | 73.84 |
| Leu 177 | 179 Asn | O … HD21-ND2 | 67.49 | Ser 125 | 266 Ser | O … H-N | 62.39 |
| Asp 174 | 165 Arg | OD2 … HH*-NH* | 66.19 | Asp 185 | 165 Arg | OD1 … H*-N* | 55.27 |
| Ser 163 | 165 Arg | O … HE-NE | 61.19 | Asp 174 | 136 Lys | OD2 … HZ*-NZ | 51.10 |
| Asp 174 | 165 Arg | OD1 … HH*-NH* | 61.10 | Asn 267 | 171 Arg | O … HH*-NH* | 46.94 |
| Gln 184 | 227 Arg | OE1 … HH12-NH1 | 59.77 | Gln 182 | 171 Arg | O … HH*-NH2 | 46.52 |
| Asp 185 | 145 Lys | OD1 … HZ*-NZ | 55.37 | Asp 185 | 165 Arg | OD2 … H*-N* | 43.98 |
| Asp 174 | 176 Lys | OD1 … HZ*-NZ | 39.71 | Gln 164 | 228 Ala | NE2-HE21 … O | 42.95 |
| Gln 182 | 179 Asn | OE1 … H*-N* | 37.73 | Met 162 | 136 Lys | O … HZ*-NZ | 34.79 |
| Cys 175 | 176 Lys | O … HZ*-NZ | 33.73 | Ser 266 | 171 Arg | O … HH*-NH1 | 32.29 |
| Asp 174 | 176 Lys | OD2 … HZ*-NZ | 33.60 | Gln 164 | 256 His | OE1 … HE2-NE2 | 30.74 |
| Arg 227 | 261 Glu | NH2-HH* … OE* | 28.51 | | | | |
| **hydrophobic interactions** | | | | **hydrophobic interactions** | | | |
| Leu 177 | 147 Phe | CD2 … CD2 | 88.36 | Phe 127 | 229 Phe | CZ … CD2 | 92.20 |
| Met 162 | 109 Leu | CB … CD2 | 84.51 | Trp 172 | 228 Ala | CZ3 … CB | 90.93 |
| Leu 177 | 147 Phe | CD1 … CD1 | 73.50 | Leu 177 | 147 Phe | CD1 … CD2 | 77.41 |
| Leu 177 | 147 Phe | CD1 … CD2 | 68.80 | Leu 177 | 109 Leu | CD1 … CD2 | 55.16 |
| Leu 177 | 149 Phe | CD2 … CZ | 56.31 | Leu 177 | 147 Phe | CD1 … CD1 | 38.18 |
| Pro 262 | 225 Pro | CD … CD | 54.98 | Val 159 | 149 Phe | CG1 … CE1 | 27.95 |
| Leu 177 | 147 Phe | CD1 … CZ | 42.82 | Leu 177 | 109 Leu | CD1 … CD1 | 25.86 |
| Met 162 | 109 Leu | CE … CD1 | 33.71 | Val 159 | 149 Phe | CG1 … CE1 | 23.79 |
| Leu 177 | 109 Leu | CD1 … CD1 | 30.98 | | | | |
| Leu 177 | 147 Phe | CD2 … CD1 | 20.75 | | | | |

| $S_\beta S_\beta$/D | | occupancy% | | $S_\beta H_{1\alpha}2$/D | | occupancy% | |
|---|---|---|---|---|---|---|---|
| **hydrogen bonds** | | | | **hydrogen bonds** | | | |
| Arg 165TRP172ARG165TYR155 Asp 185 | 170 Gly | N-H … O | 56.93 | Arg 165 | 259 Asn | NH2-HH22 … O | 62.26 |

(Continued.)

**Table 2.** (*Continued.*)

| S$_\alpha$S$_\beta$/D | | | occupancy% | S$_\beta$H$_{1\alpha}$2/D | | | occupancy% |
|---|---|---|---|---|---|---|---|
| Arg 171 | NH*-HH*…O | 255 Ile | 39.28 | Arg 171 | NE1-HE1…O | Trp 172 | 52.92 |
| Glu 186 | OE*…H*-N* | 171 Arg | 32.04 | Glu 186 | NH2-HH21…OG | Arg 165 | 51.93 |
| Ser 183 | O…H-N | 172 Trp | 30.28 | Ser 183 | OH-HH…O | Tyr 155 | 23.63 |
| Ala 228 | N-H…O | 141 Thr | 26.26 | | **hydrophobic interactions** | | |
| Asn 179 | ND2-HD21…O | 172 Trp | 25.77 | Asn 179 | CG1…CD2 | Val 159 | 79.74 |
| Arg 171 | NH1-HH1…O | 246 Glu | 24.71 | Arg 171 | CG1…CD1 | Val 159 | 73.65 |
| Gln 182 | NE2-HE22…O | 172 Trp | 23.45 | Gln 182 | CG1…CD2 | Val 159 | 71.89 |
| Asn 259 | O…HH*-NH* | 171 Arg | 23.00 | Asn 259 | CG2…CD1 | Val 159 | 61.31 |
| Arg 171 | NH1-HH1…O | 178 Pro | 20.57 | Arg 171 | CE…CD1 | Met 167 | 50.17 |
| **hydrophobic interactions** | | | | | CE…CD2 | Met 167 | 47.86 |
| Phe 147 | CD1…CZ | 147 Phe | 94.88 | Phe 147 | CD1…CB | Trp 172 | 48.54 |
| Trp 113 | CD1…CZ | 194 Phe | 90.63 | Trp 113 | CD1…CD | Trp 172 | 36.94 |
| Trp 113 | CD1…CD1 | 194 Phe | 71.93 | Trp 113 | CG1…CZ | Val 159 | 36.02 |
| Trp 113 | CD1…CD2 | 194 Phe | 70.50 | Trp 113 | CG2…CD2 | Val 159 | 34.38 |
| Phe 147 | CD2…CZ | 147 Phe | 64.14 | Phe 147 | CG1…CD2 | Val 159 | 26.98 |
| Leu 109 | CD2…CZ | 147 Phe | 49.35 | Leu 109 | CG2…CZ | Val 159 | 25.94 |
| Phe 147 | CZ…CZ | 147 Phe | 46.07 | **H$_1$H$_1$/D** | | | |
| Trp 113 | CE3…CZ | 229 Phe | 45.86 | **hydrogen bonds** | | | |
| Trp 113 | CZ2…CD1 | 194 Phe | 33.55 | Phe 147 | O…HE21-NE2 | 167 Met | 70.14 |
| Trp 113 | CE3…CZ | 194 Phe | 27.96 | Trp 113 | O…H-N | 168 Ile | 65.99 |
| Leu 109 | CD2…CD1 | 147 Phe | 23.49 | Leu 109 | NZ…HZ*-O | 114 Lys | 59.12 |
| Leu 109 | CD1…CD2 | 109 Leu | 20.45 | Leu 109 | OE*…HH22-NH2 | 122 Glu | 20.71 |
| | | | | **hydrophobic interactions** | | | |
| | | | | 168 Ile | CG2…CH2 | 172 Trp | 80.31 |
| | | | | 123 Tyr | CD2…CH2 | 172 Trp | 32.52 |

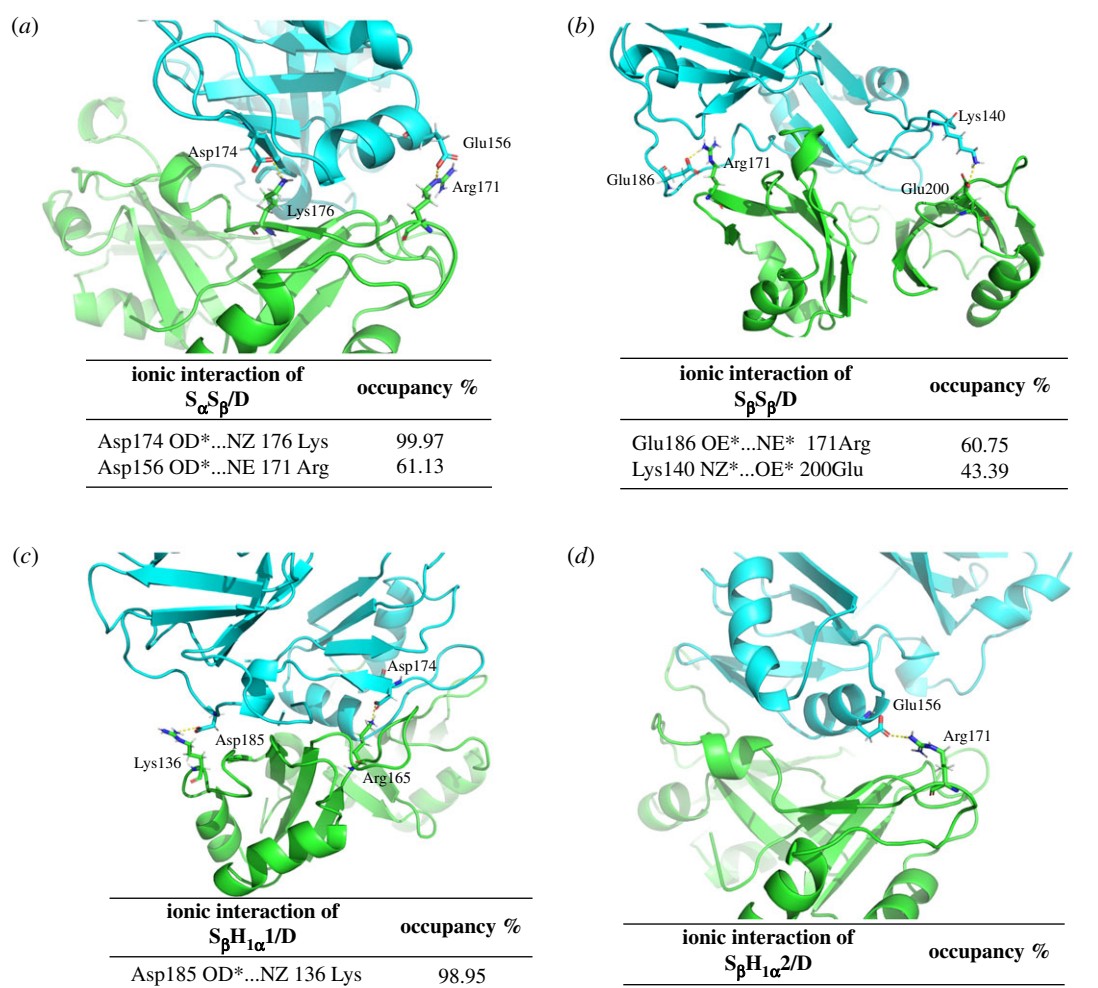

**Figure 4.** The ionic interaction occupancies between two monomer RRM12 from the simulations of the dimer models of (a) $S_\alpha S_\beta$/D, (b) $S_\beta S_\beta$/D, (c) $S_\beta H_{1\alpha}1$/D and (d) $S_\beta H_{1\alpha}2$/D, respectively.

## 3.3. The oligomer aggregation mechanism

### 3.3.1. The structure and interaction characteristics of tetramer models

To investigate the aggregation mechanism of RRM1 and RRM2 domains of TDP-43 protein, two tetramer models of $S_\alpha S_\beta 1$/T and $S_\alpha S_\beta 2$/T based on the most stable $S_\alpha S_\beta$/D dimer model were built by using the ZDOCK software with the top energy scoring, and simulated for 50 ns by using the same method described in §2.2. These two tetramer models represent the typical aggregation modes among the first eight docking structures with the top scores from the 250 docking models screened by the ZDOCK software. The equilibrated conformations of $S_\alpha S_\beta 1$/T and $S_\alpha S_\beta 2$/T are shown in figure 6a and b, and RMSD values are shown in electronic supplementary material, figure S1f. It can be seen from figure 6a that the $S_\alpha S_\beta 1$/T tetramer model presents that both head-1 sides of RRM1 domains in the cyan and green $S_\alpha S_\beta$/D dimer model combine to both head-1 sides of RRM1 domains in the yellow and magenta $S_\alpha S_\beta$/D dimer model, i.e. two dimers are bound together in the way of head-to-head along the shoulder horizontal direction. However, the $S_\alpha S_\beta 2$/T tetramer model in figure 6b presents that both head-2 sides of RRM2 domains in the cyan and green $S_\alpha S_\beta$/D dimer model combine with both head-2 sides of RRM2 domains in the yellow and magenta $S_\alpha S_\beta$/D dimer model in the direction of head-to-head opposite to the first dimer, i.e. head-2 to head-2 along the tilt head direction. The binding free energies (shown in electronic supplementary material, tables S1 and S2) in the head-1-to-head-1 or head-2-to-head-2 interfaces between two $S_\alpha S_\beta$/D dimer models for $S_\alpha S_\beta 1$/T and $S_\alpha S_\beta 2$/T tetramer models are, respectively, −17.61 and −32.05 kcal mol$^{-1}$; meanwhile, the average binding free energies of −54.47 and −53.66 kcal mol$^{-1}$ in the shoulder-to-shoulder interfaces of $S_\alpha S_\beta$/D dimer (in

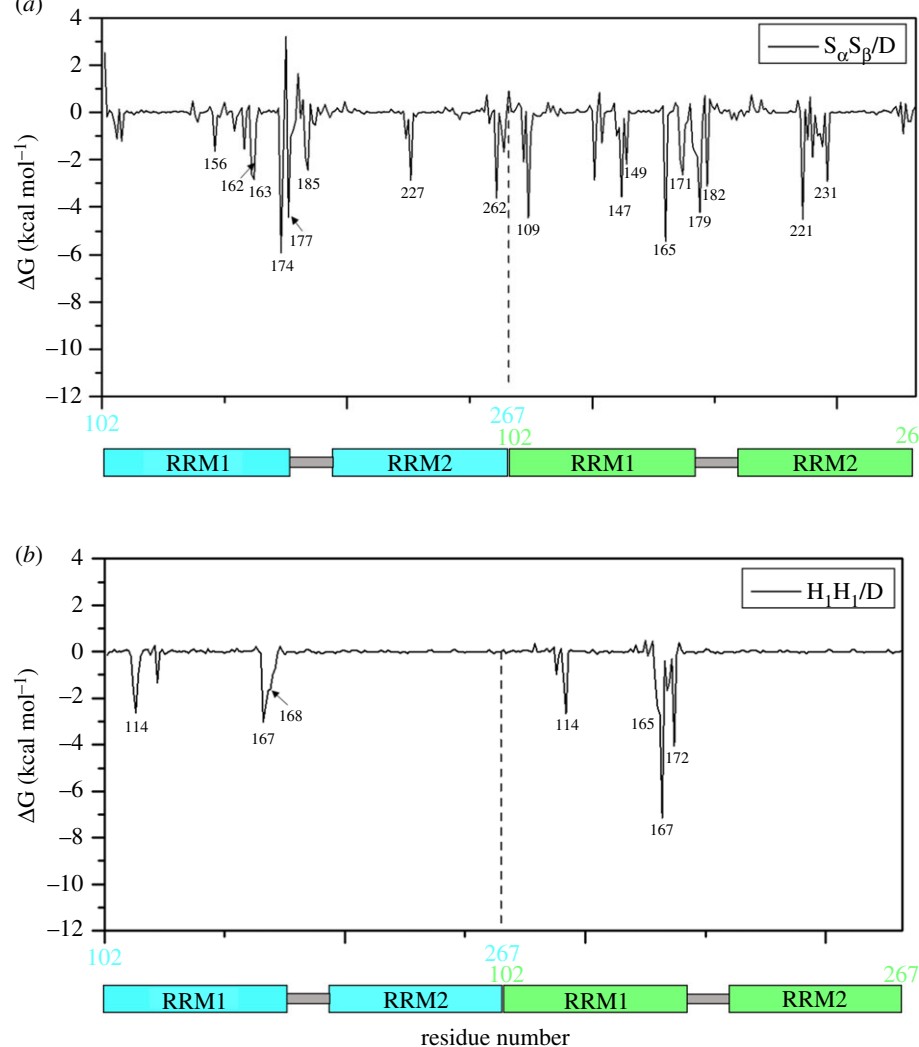

**Figure 5.** MM-PBSA energy decompositions between two monomer RRM12 representing (kcal mol$^{-1}$) into the residues of RRM12 domains for the dimer (*a*) S$_\alpha$S$_\beta$/D and (*b*) H$_1$H$_1$/D models.

figure 6*a*) in the tetramer S$_\alpha$S$_\beta$1/T and S$_\alpha$S$_\beta$2/T models, respectively, are similar to the binding free energy of −57.66 kcal mol$^{-1}$ in the same interface of the S$_\alpha$S$_\beta$/D model, which suggests that the affinity of shoulder-to-shoulder interface in the tetramer models presents the consistent stability compared with the corresponding dimer model, which would favour the formation of tetramer aggregation. The details of others tetramer models built from the S$_\beta$S$_\beta$/D and S$_\beta$H$_{1\alpha}$1/D dimer models are also analysed, and are shown in section S3, figure S1f and S3 of the electronic supplementary material.

To investigate the relationship between structural dynamics and interactions, the dynamical fluctuations of every residue of one same RRM1 domain in the RRM12, S$_\alpha$S$_\beta$/D and S$_\alpha$S$_\beta$1/T models were determined by RMSF values (figure 7*a*). It can be observed that the RMSF values in the dimer S$_\alpha$S$_\beta$/D model and tetramer S$_\alpha$S$_\beta$1/T model are smaller than those in the monomer RRM12 model, which is usually considered directly linking to the formations of new hydrogen bonds, hydrophobic and ionic interactions, and further supports the interaction analysis (table 2 and figure 4; electronic supplementary material, table S2). Moreover, due to the interactions for the dimer S$_\alpha$S$_\beta$/D model and the tetramer S$_\alpha$S$_\beta$1/T models mainly achieved by the parallel β-sheet layers of RRM1 domains and the connected flexible loops, as expected, the residues in β1, β4 and β5 of parallel β-sheet layer of one RRM1 domain have the lower RMSF values. To address further the stability of affinity at the shoulder-to-shoulder interface in the tetramer S$_\alpha$S$_\beta$1/T model, the motion correlations of the allosteric communication for the backbone atoms during the first 10 ns simulation time in the aggregation process from the dimer S$_\alpha$S$_\beta$/D model to tetramer S$_\alpha$S$_\beta$1/T model have been calculated, and the corresponding cross correlation map is shown in figure 7*b*. In this figure, the range of the residue motion correlations is labelled from low anticorrelation (blue) to high correlations (red). It can be

(*a*)    $S_\alpha S_\beta 1/T$

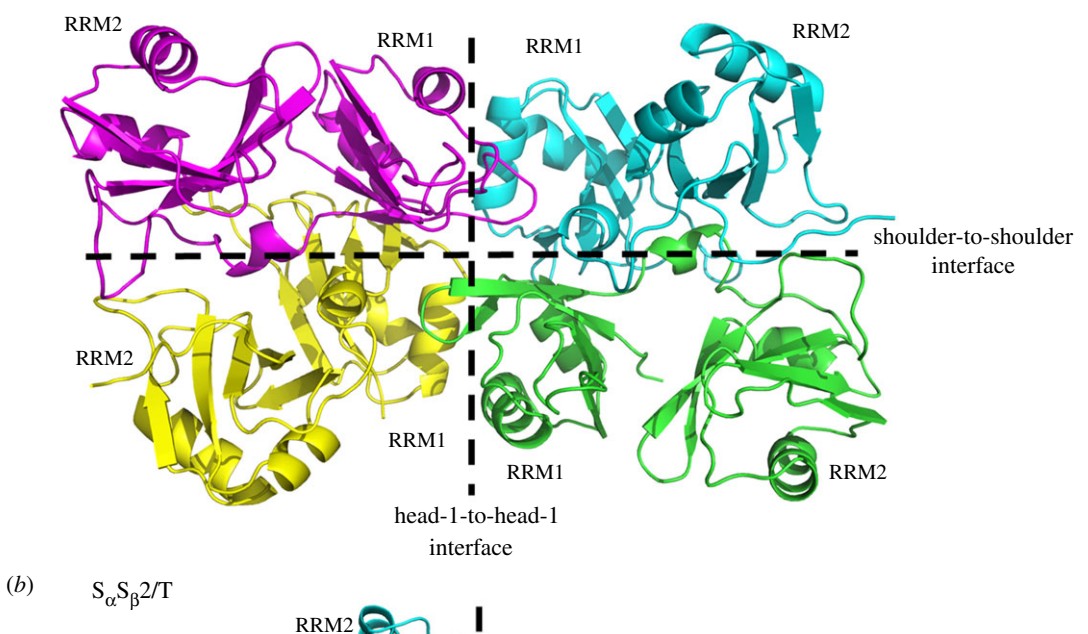

(*b*)    $S_\alpha S_\beta 2/T$

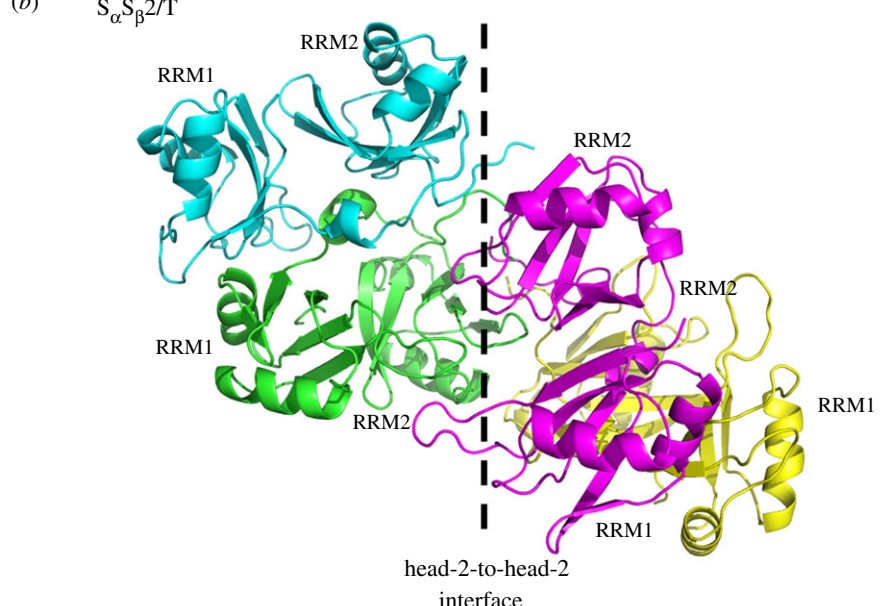

**Figure 6.** Average structures of the tetramer models of (*a*) $S_\alpha S_\beta 1/T$ and (*b*) $S_\alpha S_\beta 2/T$. The shoulder-to-shoulder, head-1-to-head-1 and head-2-to-head-2 interfaces are marked by dash lines.

found that the motions of α1 and α2 helices of the RRM1 domains in the head-to-head interface significantly correlate to the motions of the parallel β-sheet layer (β1, β4 and β5) of the same RRM1 domains in the shoulder-to-shoulder interface, which implies the allosteric communication in the RRM1 domains from the α1 and α2 helices in the head-to-head interface to the parallel β-sheet layer in the shoulder-to-shoulder interface.

Moreover, the binding free energy contributed from the RRM1 domains for the $S_\alpha S_\beta 1/T$ tetramer model is more favourable than that from the RRM2 domains for the $S_\alpha S_\beta 1/T$ tetramer model, which is consistent with the calculations of binding free energies for the dimer models, also suggests that the aggregation coming from the RRM1 domains is dominant, and that from the RRM2 domains is the second choice. For this reason, it is found that the residues of Leu109 of β1, Asp165 of β4, Arg171, Asp174, Lys176, Leu177 of β5 on the β-sheet layers in the RRM1 domain might be the main contribution to the oligomer aggregation; accordingly, these residues are absent in the RRM2 domain for the $S_\alpha S_\beta/D$ model. This result is consistent with the experimental fact that the β-sheet layers of the RRM1 domain is an important motif with the highly conserved sequence for RNA binding ability and the pathological of TDP-43 aggregation [57].

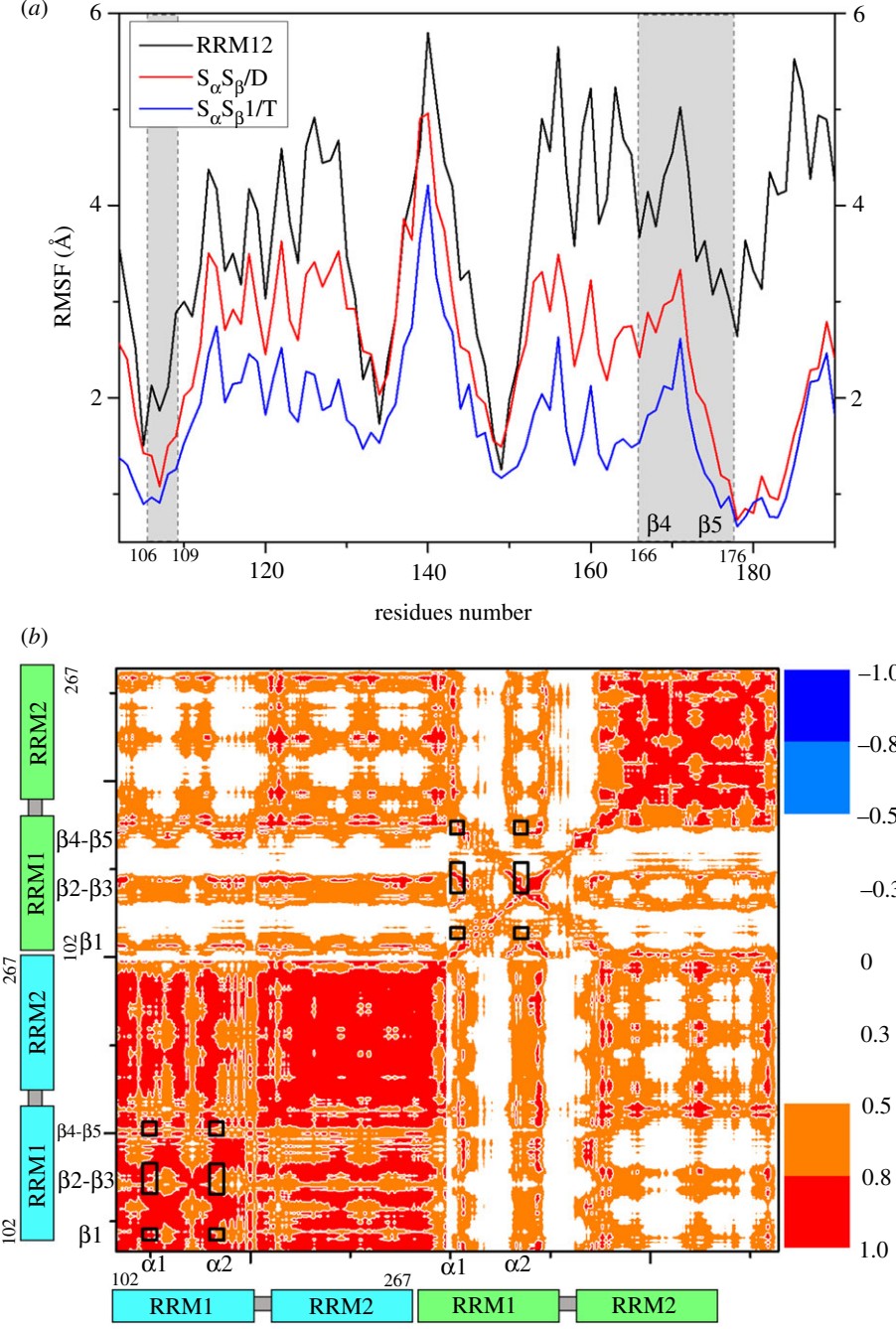

**Figure 7.** (a) The residues fluctuations of RRM1 domain for the RRM12 (black line), $S_\alpha S_\beta/D$ (red line) and $S_\alpha S_\beta 1/T$ (blue line) models. (b) Dynamical cross-correlation maps calculated from the first 10 ns of the $S_\alpha S_\beta 1/T$ model for the residues of the $S_\alpha S_\beta/D$ dimer (yellow and magenta in figure 6a). The key subregions are marked by squared in black.

### 3.3.2. The mechanism of the large oligomer aggregations

To explore the aggregation mechanism and to explain the pathogenic aggregation phenomenal *in vivo* [33], we try to construct the large aggregation models with the periodic structures from the small aggregation models. It is obvious that the $S_\alpha S_\beta 1/T$ and $S_\alpha S_\beta 2/T$ tetramer models present the similar structures besides of the different combining directions of head-1-to-head-1 or head-2-to-head-2 from two $S_\alpha S_\beta/D$ dimer models. Especially, each $S_\alpha S_\beta/D$ dimer in the two tetramers still shows the similar conformations to the corresponding isolate $S_\alpha S_\beta/D$ dimer. To construct further the large aggregation models, the superimposing-creating pattern of the hexamer model (as called $S_\alpha S_\beta/H$ model) from the $S_\alpha S_\beta 1/T$ and $S_\alpha S_\beta 2/T$ tetramer models is shown in figure 8 with the scheme of model superimposing

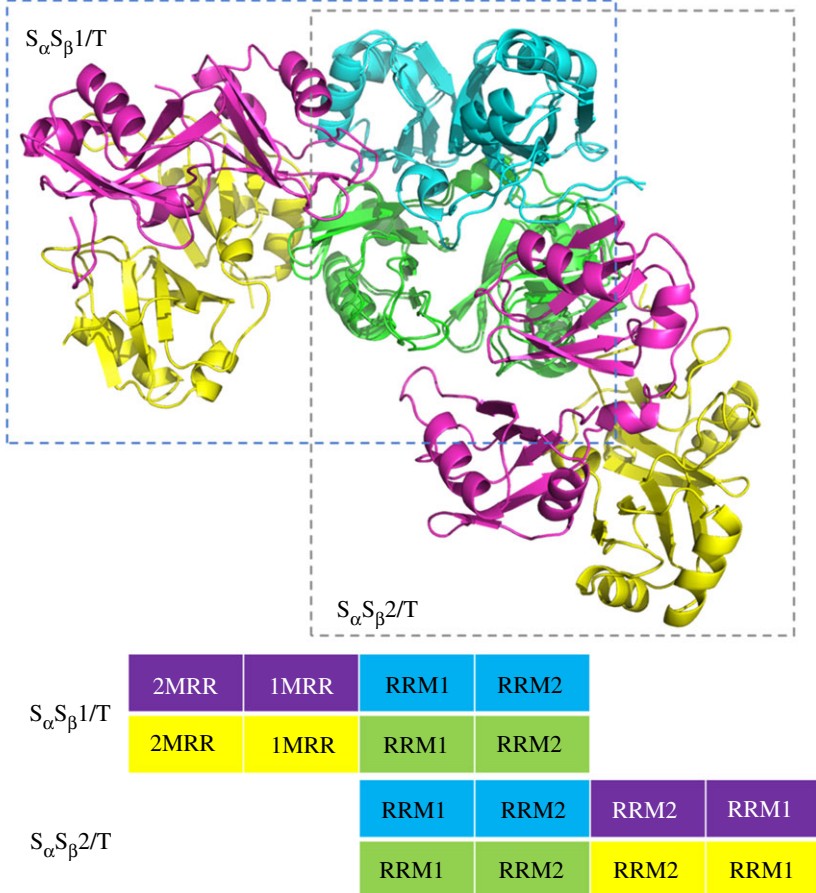

| $S_\alpha S_\beta 1/T$ | 2MRR | 1MRR | RRM1 | RRM2 | | |
|---|---|---|---|---|---|---|
| | 2MRR | 1MRR | RRM1 | RRM2 | | |
| $S_\alpha S_\beta 2/T$ | | | RRM1 | RRM2 | RRM2 | RRM1 |
| | | | RRM1 | RRM2 | RRM2 | RRM1 |

**Figure 8.** The hexamer model of $S_\alpha S_\beta$/H was constructed using the superimposing technique from the $S_\alpha S_\beta 1$/T and $S_\alpha S_\beta 2$/T tetramer models. The scheme of $S_\alpha S_\beta$/H model superimposing by using firstly an $S_\alpha S_\beta$/D dimer of the $S_\alpha S_\beta 2$/T tetramer model overlapping on the $S_\alpha S_\beta 1$/T model, then, by removing the $S_\alpha S_\beta$/D dimer with the superimposing technique of the PyMol program (http://www.pymol.org).

by firstly using an $S_\alpha S_\beta$/D dimer of the $S_\alpha S_\beta 2$/T tetramer model overlapping on the $S_\alpha S_\beta 1$/T model, and then removing the repetitive $S_\alpha S_\beta$/D dimer with the superimposing technique of the PyMol program (http://www.pymol.org). Based on this superimposing method, the two possible paths for establishing the periodic octamer model from the $S_\alpha S_\beta 1$/T and $S_\alpha S_\beta 2$/T tetramer models are shown in figure 9a, in which an octamer model can be obtained along two different overlapping directions on the $S_\alpha S_\beta 1$/T tetramer model, i.e. by using the left $S_\alpha S_\beta$/D dimer of the $S_\alpha S_\beta 2$/T tetramer model overlapping on the right $S_\alpha S_\beta$/D dimer of the $S_\alpha S_\beta 1$/T tetramer model (path 1 in figure 9a) or by using the right $S_\alpha S_\beta$/D dimer overlapping on the left $S_\alpha S_\beta$/D dimer (path 2 in figure 9a). The two octamer models constructed from these two paths are shown in figure 9b, and present full equal conformations with the building unit as the same as the $S_\alpha S_\beta 1$/T tetramer model. Consequently, the large aggregation models could be established by using the similar constructing pathways. For example, the dodecamer model constructed by using the same repeatable-superimposing method is shown in figure 9c. Interestingly, the structure of the dodecamer model exhibits the appearance of repeatable, helical and rope-like aggregation pattern which may imply the structural characteristics of the protein aggregation as observed in experiment [30].

Furthermore, the $S_\alpha S_\beta$/H hexamer model constructed by path 1 was simulated for 50 ns by using the same method described in §2.2, and the equilibrated conformation is shown in electronic supplementary material, figure S4. The small RMSD values analysed from the trajectory of this model support the stable existence possibility of this hexamer model, and are shown in electronic supplementary material, figure S1 g. The average binding free energy for two head-to-head interface in the hexamer $S_\alpha S_\beta$/H model are −18.2 kcal mol$^{-1}$. Moreover, the average binding free energy of −54.20 kcal mol$^{-1}$ for the shoulder-to-shoulder interface of hexamer $S_\alpha S_\beta$/H model is similar to that of −54.47 kcal mol$^{-1}$ for that of tetramer $S_\alpha S_\beta 1$/T model, which suggests that the affinity of shoulder-to-shoulder interface in hexamer $S_\alpha S_\beta$/H model also maintains the consistent stability by the formation of new head-1-to-head-1 and

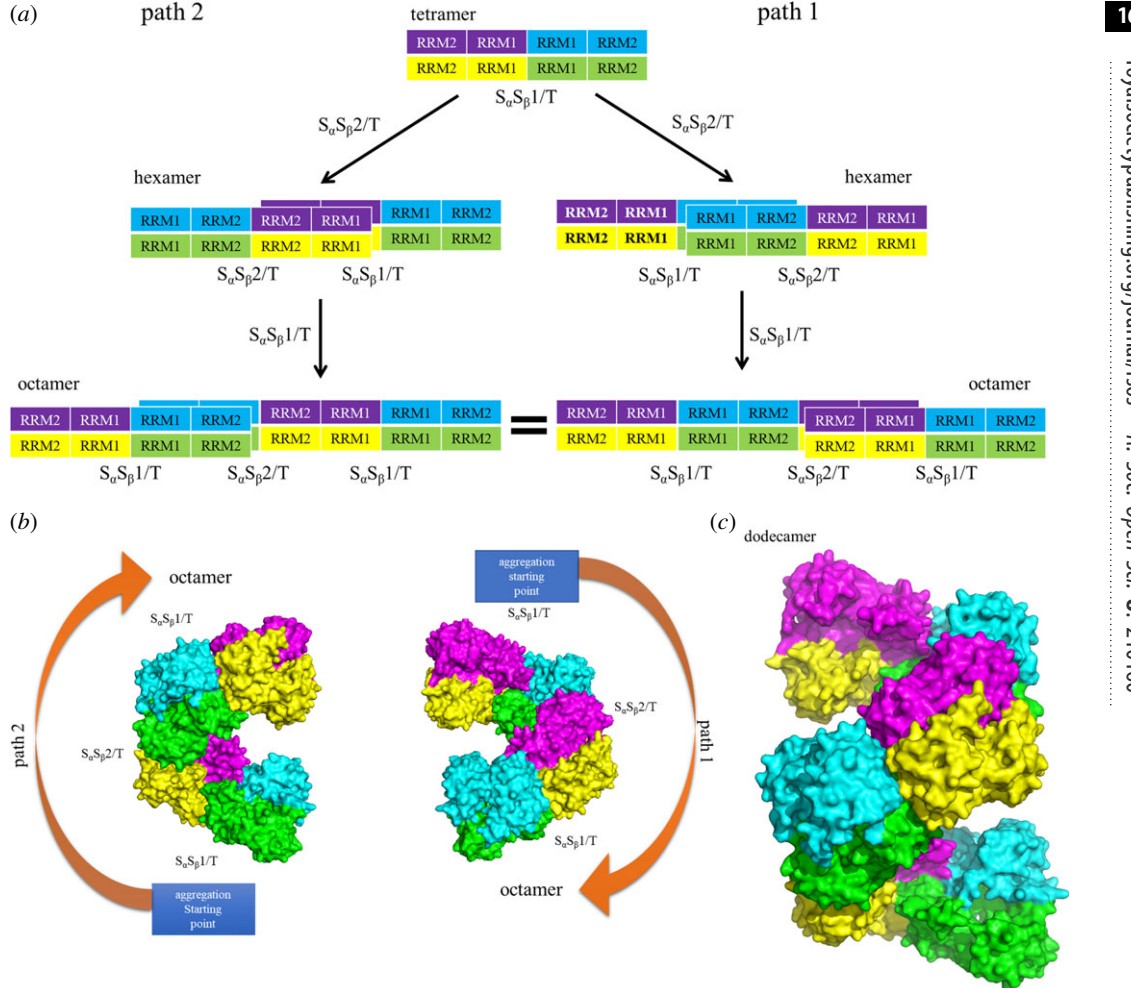

**Figure 9.** (*a*) The two possible paths for establishing the periodic octamer model from the $S_\alpha S_\beta 1/T$ and $S_\alpha S_\beta 2/T$ tetramer models were depicted, i.e. by using the left $S_\alpha S_\beta/D$ dimer of the $S_\alpha S_\beta 2/T$ tetramer model overlapping on the right $S_\alpha S_\beta/D$ dimer of the $S_\alpha S_\beta 1/T$ tetramer model (path 1) or by using the right $S_\alpha S_\beta/D$ dimer of that overlapping on the left $S_\alpha S_\beta/D$ dimer (path 2). (*b*) Representation of two octamer models constructed from two possible paths. (*c*) The dodecamer model was constructed using the same repeatable-superimposing method.

head-2-to-head-2 interfaces. This result indicates that there is the concerted interaction affinity between the shoulder-to-shoulder and head-to-head interfaces. However, since it is the head-to-head interaction that contributes to the growth of the polymerization, it is expected this concerted interaction affinity will reach a constant as the number of the head-to-head interfaces produced by the aggregation process becomes large enough. The details of binding free energies were shown in electronic supplementary material, table S3. These results revealed the stability existence possibility of the hexamer model, that is, the possible aggregation process from tetramer to large oligomer aggregation using the repeatable-superimposing method is feasible, and may produce a repeatable, helical and rope-like structure as a mimicked aggregation mode.

# 4. Discussion

## 4.1. Analyses of dimer structures

To investigate the possible aggregation mechanism of small oligomer models about RRM1 and RRM2 domains, the structural characteristics and interactions of the dimer $S_\alpha S_\beta/D$, $S_\beta S_\beta/D$, $S_\beta H_{1\alpha}1/D$, $S_\beta H_{1\alpha}2/D$ and $H_1 H_1/D$ models were further elucidated. As the energy decomposition and interaction analyses discussed above, the dimerization for the dimer $S_\alpha S_\beta/D$, $S_\beta S_\beta/D$, $S_\beta H_{1\alpha}1/D$ models is mainly

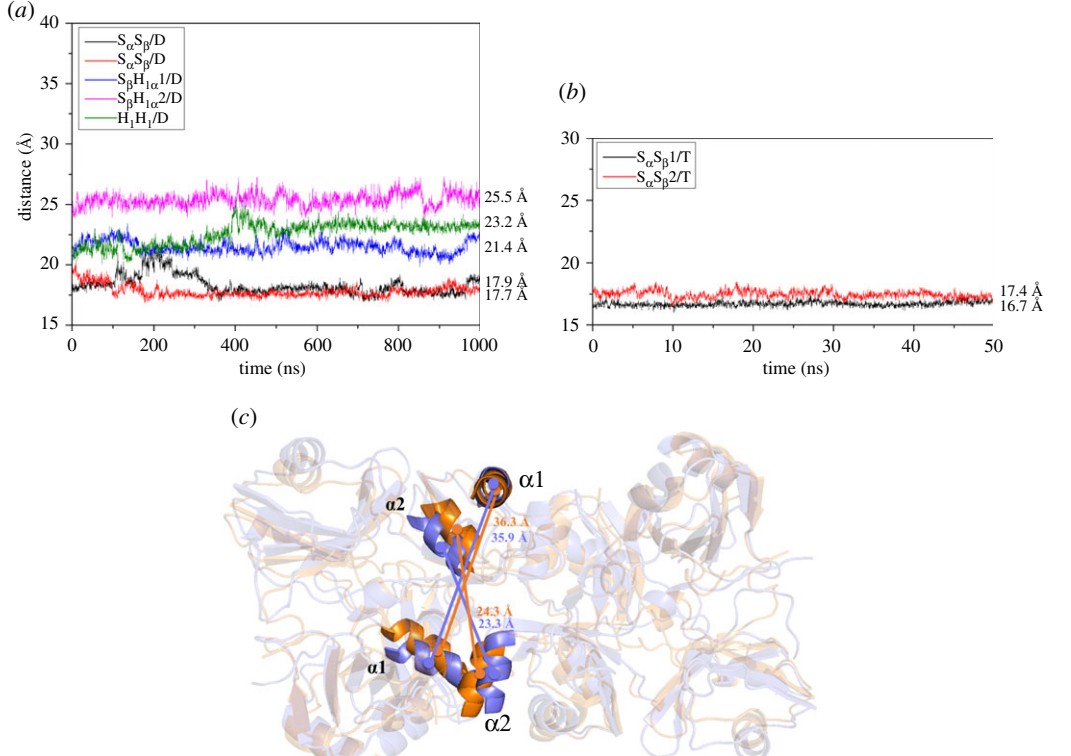

**Figure 10.** (a) Distances of time-dependences between two β-sheet layers in two RRM1 domains for the dimer $S_\alpha S_\beta$/D (black), $S_\beta S_\beta$/D (red), $S_\beta H_{1\alpha}1$/D (blue), $S_\beta H_{1\alpha}2$/D (magenta), $H_1 H_1$/D (green) models and (b) those for the $S_\alpha S_\beta$/D dimer (yellow and magenta in figure 6a) in the tetramer $S_\alpha S_\beta 1$/T (black) and $S_\alpha S_\beta 2$/T (red) models; (c) the distances of two α1 in both RRM1 domains, and those of two α2 in both RRM1 domains for the dimer $S_\alpha S_\beta$/D model (orange) and the tetramer $S_\alpha S_\beta 1$/T model (light blue).

achieved by the parallel β-sheet layers of two RRM1 domains and the connected flexible loops, but RRM2 domains in these dimer models present the weaker interactions and affinities for the dimerization. For example, the interactions and affinities in the $S_\alpha S_\beta$/D model occur mainly at Leu109 of β1; Phe147, Phe149 of β3; Glu156, Met162, Ser163 of α2; Asp165 of β4; Arg171, Asp174, Lys176, Leu177 of β5 in RRM1, which are associated with the interaction area of TDP-43 for RNA binding in experimental results [17,18]. Especially, β1, β4 and β5 of two RRM1 domains form the typical interacting surface on the parallel β-sheet layers for RNA binding. The similar results can be obtained in the dimer $S_\beta S_\beta$/D and $S_\beta H_{1\alpha}1$/D models that are shown in electronic supplementary material, table S4. It can be seen that the binding sites located at the parallel β-sheet layers for the dimerization of RRM1 in TDP-43 are almost in the same positions for RNA binding domains [18], which suggest that TDP-43 firstly binding to RNA will prevent or slow down TDP-43 aggregations due to the same occupied binding sites. These results imply that the parallel β-sheet layer of two RRM1 domains plays an important role in the protein dimerization, and RNA binding can protect the pathogenic aggregation of TDP-43, which are consistent with experimental results [33,58,59]. In addition, the parallel β-sheet layers as the feature of amyloid-like fibril structure is universally accepted [60], so the TDP-43 protein might form the large aggregation structure on basis of the interaction formations of the parallel β-sheet layer of two RRM1 domains.

Moreover, the distances between two β-sheet layers in two RRM1 domains for the $S_\alpha S_\beta$/D, $S_\beta S_\beta$/D and $S_\beta H_{1\alpha}1$/D models are 17.9, 17.7 and 21.4 Å (figure 10a), and the two β-sheet layers are parallel to each other, which might favour to form the parallel β-sheet structure as the character of amyloid-like fibril structure. Otherwise, the distances for the $S_\beta H_{1\alpha}2$/D and $H_1 H_1$/D models are 25.5 and 23.2 Å (figure 10a), respectively, and the two β-sheet layers of two RRM1 domains are not parallel in the two models (figure 3d and e), which indicates that the parallel β-sheet structure cannot be formed due to the corresponding long distances of the two β-sheet layers and their structural arrangement. That is, the $S_\alpha S_\beta$/D, $S_\beta S_\beta$/D and $S_\beta H_{1\alpha}1$/D models might possibly form the large oligomer aggregation. Moreover, the secondary structures were changed during the dimerization from the monomer RRM12

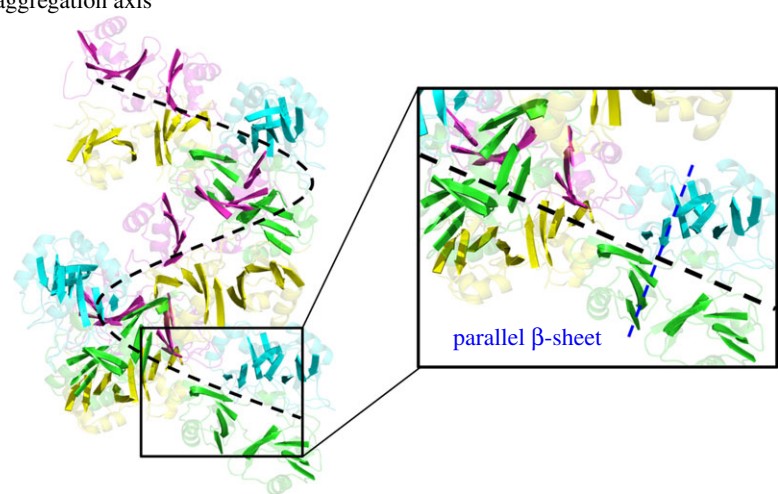

aggregation axis

parallel β-sheet

**Figure 11.** The aggregation axis (black line) existed in the large oligomer model. The inset represents that the parallel β-sheet layer orients perpendicularly to the aggregation axis.

model to some dimer models. For example, it is found through comparing the average structures of the RRM1 and RRM2 domains in the monomer RRM12 with the dimer $S_\beta S_\beta$/D model (see figures 2*b* and 3*b*) that the loose β-strand link between α2 and β5 in monomer forms the stable β4-strand (from His166 to Ile168) and the β5 length of RRM1 domain is increased from three residues (from Asp174 to Lys176) in monomer to five residues (from Arg171 to Cys176) in the $S_\beta S_\beta$/D model (green in figure 3*b*). However, the length of α2 helix with seven residues (from Asp237 to Leu243) in RRM2 domain of monomer RRM12 decreases to that with four residues (from Ala240 to Leu243); the α-helix linking loop between the RRM1 and RRM2 domains in monomer RRM12 model changes to the loop linking conformation in the same dimer $S_\beta S_\beta$/D model.

## 4.2. Analyses of the oligomer aggregation

To elucidate further the possible mechanism of large aggregation through addressing the formation progress from small oligomer to large oligomer, the oligomer models about RRM1 and RRM2 domains were built on the basis of the periodic aggregation mode from the dimer models. The amyloid aggregation possesses a cross-β structure, in which β-strands orient perpendicularly to the aggregation axis (figure 11), and are assembled into β-sheet layers that run the length of the aggregation, as detected by experiment results [37,38,60]. In the tetramer models of $S_\alpha S_\beta 1$/T and $S_\alpha S_\beta 2$/T, the interactions in the shoulder-to-shoulder interface mainly occurred at Leu109 of β1; Phe147, Phe149 of β3; Glu156, Met162, Ser163 of α2; Asp165 of β4; Arg171, Asp174, Lys176, Leu177 of β5 in RRM1, which are almost similar to those in the $S_\alpha S_\beta$/D dimer model. The details of the interactions in shoulder-to-shoulder interface for $S_\alpha S_\beta 1$/T, $S_\alpha S_\beta 2$/T models are shown in electronic supplementary material, table S5. As expected, the residues in β1, β4 and β5 of the β-sheet layer of RRM1 domain in the tetramer $S_\alpha S_\beta 1$/T model have the lower RMSF values compared with those in the $S_\alpha S_\beta$/D model. Further, the distances between the two β-sheet layers of two RRM1 domains for the $S_\alpha S_\beta$/D dimer (yellow and magenta in figure 6*a*) in the $S_\alpha S_\beta 1$/T and $S_\alpha S_\beta 2$/T models are, respectively, 16.7 and 17.4 Å (figure 10*b*), which are smaller than that of 17.9 Å in the $S_\alpha S_\beta$/D model discussed above. The approaching of the parallel β-sheet layers induces energetically to the consistent interaction affinity in the shoulder-to-shoulder interface for the tetramer model as discussed in §3.3.1. As discussed in the correlation analyses in figure 7, α1 and α2 helices of the RRM1 domains in the head-to-head interface correlates to the parallel β-sheet layers (including β1, β4 and β5) of the same RRM1 domains in the shoulder-to-shoulder interface with large cross-correlate coefficients, which represents the allosteric communication from α1 and α2 to the parallel β-sheet layers (i.e. β1, β4 and β5). Such communications relate to the structural changes around α1, α2 and the parallel β-sheet layers. Namely, the tetramer aggregation caused the movement of α1 and α2 in the head-to-head interface of the $S_\alpha S_\beta$/D dimer, i.e. the distances of two α1 in both RRM1 domains and two α2 in both RRM1 domains of the $S_\alpha S_\beta$/D dimer (yellow and magenta in figure 6*a*) decrease from 36.3 to 35.9 Å

and from 24.3 to 23.3 Å, respectively, as shown in figure 10*c*. Because α1 is connected to β1, and α2 is connected to β4 and β5, the movements of α1 and α2 relate definitely to the movements of the parallel β-sheet layers, as expected in the correlations of α1 and α2 to the parallel β-sheet layers. The approaching of the parallel β-sheet layers indicates that the parallel β-sheet layers of RRM1 of TDP-43 in the dimer model are also retained in the corresponding tetramer model, and were perpendicular to the aggregation axis, which may favour the formation of large aggregation.

Furthermore, the large aggregation models were constructed by using the same repeatable-superimposing method described in §3.3.2, in which the structural character with the repeatable, helical and rope-like as a mimicked aggregation mode has been shown by the oligomer models. The results in the tetramer models might also meet the large aggregation models. That is, the parallel β-sheet layers still exist in the large aggregation, such as hexamer, octamer, dodecamer and so on, and orient perpendicularly to the aggregation axis; and the interactions between the parallel β-sheet layers favour the formation of the large aggregation (figure 11).

Summarily, the studied oligomer aggregation process presents two types of combination of outer edges of the two domains of TDP-43 protein, i.e. one is the shoulder-to-shoulder type which occurs at the interface between the two β-sheet layers or between one β-sheet layer and the α-helix linking loop in the different shoulder sides; the other is the head-to-head type which occurs at the interface between the two α1-α2 helix planes or between the two β-sheet layer tails in the different head sides. Consequently, the conformational changes mainly involve the tertiary structures, but the secondary structures were changed slightly in the dimer, tetramer and hexamer models. For example, the mass centre distances of 25.8, 25.8 and 25.4 Å between the RRM1 and RRM2 domains in the dimer $S_\alpha S_\beta/D$, tetramer $S_\alpha S_\beta 1/T$ and hexamer $S_\alpha S_\beta/H$ models, respectively, are longer than that of 23.9 Å in the monomer RRM12 model, which expanded the expected oligomer aggregation edges and induced the looser conformation of oligomer in the shoulder-to-shoulder interface, favouring the aggregation process.

# 5. Conclusion

We performed MD simulations, binding free energy calculations and interaction analyses of the monomers, dimers, tetramers and larger oligomers of the RRM1 and RRM2 domains in TDP-43 to reveal their aggregation mechanism and aggregation process from small oligomer to large aggregation. Among the five dimer models of TDP-43, the $S_\alpha S_\beta/D$ model was found the most stable. The dimerization for these models is mainly achieved by the interactions between the parallel β-sheet layers of two RRM1 domains. Based on the analysis of the structures, energies and the motion correlations for two tetramer models built by the most stable $S_\alpha S_\beta/D$ dimer model, it is found that the changes of positions of the α1 and α2 helixes in the head-to-head interface of the RRM1 domains in each tetramer model compared with that in the dimer model induce the approaching of the parallel β-sheet layers in the shoulder-to-shoulder interface of the same RRM1 domains, which induces energetically the consistent interaction affinity in the shoulder-to-shoulder interface. This character is still valid for the large oligomer aggregations. Using the repeatable-superimposing method based on the tetramer models, we proposed a new aggregation mechanism of RRM domains in TDP-43, which characterizes the formation of the large aggregation models with the repeated, helical and rope-like structures. This RRM domains-mediated structural characteristic may drive the fibril-like aggregation. Our results imply that the RRM domains could play an important role in TDP-43 aggregation process. It is also found that in all the models the RRM domains aggregate employing the same binding sites by which TDP-43 binds to RNA, which could explain the experimental result that the RNA binding to RNA will prevent the aggregation of TDP-43. The mimicked aggregation mode is helpful for understanding the amyloid-like aggregation mechanism of TDP-43, and providing theoretical support for ALS and FTLD disease prevention.

Data accessibility. Our data are deposited at Dryad Digital Repository: https://doi.org/10.5061/dryad.w3r2280q0. The data are provided in electronic supplementary material [61].

Authors' contributions. W.L. designed the experiments, performed the experiments and wrote the paper; W.L., J.S. and Y.W. analysed the data; C.L. and G.C. designed the experiments and reviewed the paper.

Competing interests. We declare we have no competing interests.

Funding. We received no funding for this study.

Acknowledgements. This work was supported by grants from the National Natural Science Foundation of China (nos. 22073010 and 21573020); Natural Science Foundation of Hebei province, P.R. China (grant no. B2021109004);

Colleges and Universities in Hebei Province Science and Technology Research Youth Talent Support Program (grant no. BJ2019204).

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
