## [Peer Review File · Royal Society Open Science]

Review History

RSOS-210160.R0 (Original submission)

Review form: Reviewer 1

Is the manuscript scientifically sound in its present form?

Yes

Are the interpretations and conclusions justified by the results?

Yes

Is the language acceptable?

Yes

Do you have any ethical concerns with this paper?

No

Have you any concerns about statistical analyses in this paper?

No

Recommendation?

Accept with minor revision (please list in comments)

Comments to the Author(s)

This manuscript by Liu et al. used molecular dynamics simulations and binding free energy calculations to investigate the oligomer structures of RRM domains in TDP-43 and tried to reveal the potent aggregation mechanism. A series of oligomer models had been constructed and the parallel β -sheet layers between the RRM1 domains were found to provide the potential binding sites to promote the aggregation. In addition, a new aggregation mechanism of RRM domains in TDP-43 was proposed based on the repeatable-superimposing method. Overall, the conclusions are supported by detailed simulation data. Here are several comments to be addressed:

1. In the manuscript, the outer edges of the monomer RRM12 domains were divided into four regions (i.e., shoulder-a, shoulder-b, head-1 and head-2 regions) to identify the types of different oligomer models (see figure 2b). Specifically, the shoulder- α region consisted of two α 2-helices in RRM1 and RRM2 domains, and the linking loop between RRM1 and RRM2 domains; The head-1 region consisted of α 1 and α 2 helices in RRM1 domain. According to the above definitions, the shoulder- α and head-1 regions were overlapped at α 2 helix. Why the authors chose this set of definition instead of dividing the RRM12 domains into separated regions?
2. The authors are suggested to provide some secondary structure information of the equilibrated oligomers.
3. The manuscript suggested that the aggregation coming from RRM1 domains was dominant and that from the RRM2 domain was second choice (page 15). Can the authors give some explanations based on the sequences of RRM1 and RRM2 domain?
4. Models of monomers, dimers, tetramers and larger oligomers of RRM12 domains in TDP-43 have been obtained in this manuscript. So, is there a critical oligomer size above which the oligomers are stable, like the aggregation process of human islet amyloid polypeptide (e.g., PMID: 30502402)?

Review form: Reviewer 2

Is the manuscript scientifically sound in its present form?

No

Are the interpretations and conclusions justified by the results?

No

Is the language acceptable?

Yes

Do you have any ethical concerns with this paper?

No

Have you any concerns about statistical analyses in this paper?

Yes

Recommendation?

Major revision is needed (please make suggestions in comments)

Comments to the Author(s)

Report

The manuscript entitled “Insights into the aggregation mechanism of RRM domains in TDP-43: A molecular dynamics study” by Liu et al. attempts to explore the aggregation mechanism of RRM domains in TDP43. I consider the topic very important on the disease onset of Amyotrophic lateral sclerosis (ALS) and Frontotemporal lobar degeneration (FTLD) where RRM aggregation may play a role in sclerosis pathogenesis. The authors constructed dimer, tetramer, hexamer models where I have a serious reservation about the method. The concerns are as follows:

(i) I would not consider this study as a molecular dynamics study. Instead, it is essentially docking based study using ZDOCK software to model the dimer, tetramer, etc.

(ii) The authors superpose isolate RRM1 and RRM2 to RNA-bound dimeric RRM12 structure (pdb: 4BS2) and then removed RNA. They have constructed the intermediate loops (Lys102-ASN267 and ASN179 to Val193). For this, they mentioned using NMR experimental data but didn't refer to which NMR data. After the loop reconstruction, they mutated 200Gly to 200Glu.

(iii) They adopted such extensive structural reconstruction to build the dimeric model, to begin with. Under such new changes, the system certainly demands a long-time relaxation at an atomistic level. The authors claim the equilibrium reaches immediately after 40ns and their equilibrium statistics range is only from their 40-50ns (10 ns span) for the dimer. This small timescale is unacceptable considering the timescale required for minimum structural rearrangement (of the order of μ s) of such a complex dimeric system under the new reconstruction.

(iv) For such a large system they have taken only 8 Å solvation box. In this confined boundary condition, I afraid of whether any rotation/translational movements are at all feasible to facilitate any conformational rearrangement to relax the structure.

(v) On page 8, after getting the docked structures, the authors calculate binding free energy using MM-PBSA where they mentioned that this free energy only includes enthalpy term as entropy requires much more computational resources. They also mentioned that the entropy term could be neglected when the receptor and ligand have similar conformations.

This cannot be true as the structure has a number of flexible linkers. Conformational changes are inevitable for such a hybrid structure where disordered segments would substantially be perturbed after binding to another monomer. They have deliberately ignored conformational dynamics during any sort of oligomer aggregation process.

I do not recommend this for publication unless the models are appropriately simulated using a reasonable box size for a longer time scale (at least 1 μ s because of the system's size) for relaxation and an appropriate free energy calculation at least for the dimer. Without appropriate analyses of the involved conformational changes during such oligomer aggregation process, such study will not provide any correct insight.

Decision letter (RSOS-210160.R0)

Dear Dr Li:

Title: Insights into the aggregation mechanism of RRM domains in TDP-43: A molecular dynamics study

Manuscript ID: RSOS-210160

The editor assigned to your manuscript has now received comments from reviewers. We would like you to revise your paper in accordance with the referee and Subject Editor suggestions which can be found below (not including confidential reports to the Editor). Please note this decision does not guarantee eventual acceptance.

Please submit your revised paper before 21-Apr-2021. Please note that the revision deadline will expire at 00.00am on this date. If we do not hear from you within this time then it will be assumed that the paper has been withdrawn. In exceptional circumstances, extensions may be possible if agreed with the Editorial Office in advance. We do not allow multiple rounds of revision so we urge you to make every effort to fully address all of the comments at this stage. If deemed necessary by the Editors, your manuscript will be sent back to one or more of the original reviewers for assessment. If the original reviewers are not available we may invite new reviewers.

On behalf of the Subject Editor Professor Anthony Stace and the Associate Editor Dr Debashree Ghosh.

RSC Associate Editor:
Comments to the Author:
The reviewers have raised pertinent issues which needs to be addressed by the authors.

RSC Associate Editor:
Comments to the Author:

(There are no comments.)

Reviewers' Comments to Author:

Reviewer: 1

Comments to the Author(s)

This manuscript by Liu et al. used molecular dynamics simulations and binding free energy calculations to investigate the oligomer structures of RRM domains in TDP-43 and tried to reveal the potent aggregation mechanism. A series of oligomer models had been constructed and the parallel β -sheet layers between the RRM1 domains were found to provide the potential binding sites to promote the aggregation. In addition, a new aggregation mechanism of RRM domains in TDP-43 was proposed based on the repeatable-superimposing method. Overall, the conclusions are supported by detailed simulation data. Here are several comments to be addressed:

1. In the manuscript, the outer edges of the monomer RRM12 domains were divided into four regions (i.e., shoulder-a, shoulder-b, head-1 and head-2 regions) to identify the types of different oligomer models (see figure 2b). Specifically, the shoulder- α region consisted of two α 2-helices in RRM1 and RRM2 domains, and the linking loop between RRM1 and RRM2 domains; The head-1 region consisted of α 1 and α 2 helices in RRM1 domain. According to the above definitions, the shoulder- α and head-1 regions were overlapped at α 2 helix. Why the authors chose this set of definition instead of dividing the RRM12 domains into separated regions?
2. The authors are suggested to provide some secondary structure information of the equilibrated oligomers.
3. The manuscript suggested that the aggregation coming from RRM1 domains was dominant and that from the RRM2 domain was second choice (page 15). Can the authors give some explanations based on the sequences of RRM1 and RRM2 domain?
4. Models of monomers, dimers, tetramers and larger oligomers of RRM12 domains in TDP-43 have been obtained in this manuscript. So, is there a critical oligomer size above which the oligomers are stable, like the aggregation process of human islet amyloid polypeptide (e.g., PMID: 30502402)?

Reviewer: 2

Comments to the Author(s)

Report

The manuscript entitled "Insights into the aggregation mechanism of RRM domains in TDP-43: A molecular dynamics study" by Liu et al. attempts to explore the aggregation mechanism of RRM domains in TDP43. I consider the topic very important on the disease onset of Amyotrophic lateral sclerosis (ALS) and Frontotemporal lobar degeneration (FTLD) where RRM aggregation may play a role in sclerosis pathogenesis. The authors constructed dimer, tetramer, hexamer models where I have a serious reservation about the method. The concerns are as follows:

- (i) I would not consider this study as a molecular dynamics study. Instead, it is essentially docking based study using ZDOCK software to model the dimer, tetramer, etc.
- (ii) The authors superpose isolate RRM1 and RRM2 to RNA-bound dimeric RRM12 structure (pdb: 4BS2) and then removed RNA. They have constructed the intermediate loops (Lys102-ASN267 and ASN179 to Val193). For this, they mentioned using NMR experimental data but didn't refer to which NMR data. After the loop reconstruction, they mutated 200Gly to 200Glu.
- (iii) They adopted such extensive structural reconstruction to build the dimeric model, to begin with. Under such new changes, the system certainly demands a long-time relaxation at an atomistic level. The authors claim the equilibrium reaches immediately after 40ns and their equilibrium statistics range is only from their 40-50ns (10 ns span) for the dimer. This small timescale is unacceptable considering the timescale required for minimum structural

rearrangement (of the order of μs) of such a complex dimeric system under the new reconstruction.

(iv) For such a large system they have taken only 8 Å solvation box. In this confined boundary condition, I am afraid of whether any rotation/translational movements are at all feasible to facilitate any conformational rearrangement to relax the structure.

(v) On page 8, after getting the docked structures, the authors calculate binding free energy using MM-PBSA where they mentioned that this free energy only includes enthalpy term as entropy requires much more computational resources. They also mentioned that the entropy term could be neglected when the receptor and ligand have similar conformations.

This cannot be true as the structure has a number of flexible linkers. Conformational changes are inevitable for such a hybrid structure where disordered segments would substantially be perturbed after binding to another monomer. They have deliberately ignored conformational dynamics during any sort of oligomer aggregation process.

I do not recommend this for publication unless the models are appropriately simulated using a reasonable box size for a longer time scale (at least $1\mu\text{s}$ because of the system's size) for relaxation and an appropriate free energy calculation at least for the dimer. Without appropriate analyses of the involved conformational changes during such oligomer aggregation process, such study will not provide any correct insight.

Author's Response to Decision Letter for (RSOS-210160.R0)

See Appendices A-C.

RSOS-210160.R1 (Revision)

Review form: Reviewer 1

Is the manuscript scientifically sound in its present form?

Yes

Are the interpretations and conclusions justified by the results?

Yes

Is the language acceptable?

Yes

Do you have any ethical concerns with this paper?

No

Have you any concerns about statistical analyses in this paper?

No

Recommendation?

Accept as is

Comments to the Author(s)

All comments addressed.

Review form: Reviewer 2

Is the manuscript scientifically sound in its present form?

Yes

Are the interpretations and conclusions justified by the results?

Yes

Is the language acceptable?

Yes

Do you have any ethical concerns with this paper?

No

Have you any concerns about statistical analyses in this paper?

No

Recommendation?

Accept with minor revision (please list in comments)

Comments to the Author(s)

- (i) In Figure 2A RMSD values of heavy atoms need to be updated from 50 ns to 1 μ s.
- (ii) Figure 7b, why dynamic cross has been calculated from the first 10 ns? Why not from the equilibrium trajectory extracted from 1 μ s?
- (iii) Figure 10 (a): Time-dependences of distances should be plotted for the whole 1 μ s.
- (iv) In the abstract this sentence needs be rewritten in meaningful way: "The parallel β -sheet layers between the RRM1 domains in these oligomer models, which provide the potential binding sites in the aggregation process, are formed energetically favorable."

Decision letter (RSOS-210160.R1)

Dear Dr Li:

Title: Insights into the aggregation mechanism of RRM domains in TDP-43: A theoretical exploration

Manuscript ID: RSOS-210160.R1

Thank you for submitting the above manuscript to Royal Society Open Science. On behalf of the Editors and the Royal Society of Chemistry, I am pleased to inform you that your manuscript will be accepted for publication in Royal Society Open Science subject to minor revision in accordance with the referee suggestions. Please find the reviewers' comments at the end of this email.

The reviewers and handling editors have recommended publication, but also suggest some minor revisions to your manuscript. Therefore, I invite you to respond to the comments and revise your manuscript.

Because the schedule for publication is very tight, it is a condition of publication that you submit the revised version of your manuscript before 14-Jul-2021. Please note that the revision deadline will expire at 00.00am on this date. If you do not think you will be able to meet this date please let me know immediately.

Kind regards,
Dr Laura Smith
Publishing Editor, Journals

Royal Society of Chemistry
Thomas Graham House

Science Park, Milton Road
Cambridge, CB4 0WF
Royal Society Open Science - Chemistry Editorial Office

On behalf of the Subject Editor Professor Anthony Stace and the Associate Editor Dr Debashree Ghosh.

RSC Associate Editor:

Comments to the Author:

The paper can be accepted after the authors submit the revised manuscript as requested the reviewers.

RSC Subject Editor:

Comments to the Author:

(There are no comments.)

Reviewer comments to Author:

Reviewer: 1

Comments to the Author(s)

All comments addressed.

Reviewer: 2

Comments to the Author(s)

- (i) In Figure 2A RMSD values of heavy atoms need to be updated from 50 ns to 1 μ s.
- (ii) Figure 7b, why dynamic cross has been calculated from the first 10 ns? Why not from the equilibrium trajectory extracted from 1 μ s?
- (iii) Figure 10 (a): Time-dependences of distances should be plotted for the whole 1 μ s.
- (iv) In the abstract this sentence needs be rewritten in meaningful way: "The parallel β -sheet layers between the RRM1 domains in these oligomer models, which provide the potential binding sites in the aggregation process, are formed energetically favorable."

Author's Response to Decision Letter for (RSOS-210160.R1)

See Appendix D.

Decision letter (RSOS-210160.R2)

Dear Dr Li:

Title: Insights into the aggregation mechanism of RRM domains in TDP-43: A theoretical exploration

Manuscript ID: RSOS-210160.R2

It is a pleasure to accept your manuscript in its current form for publication in Royal Society Open Science. The chemistry content of Royal Society Open Science is published in collaboration with the Royal Society of Chemistry.

On behalf of the Subject Editor Professor Anthony Stace and the Associate Editor Dr Debashree Ghosh.

RSC Associate Editor

Comments to the Author:

The authors have addressed all the issues and clarifications sought by the referees and therefore, I suggest that the manuscript be accepted as is.

Reviewer(s)' Comments to Author:

Appendix A

Reply letter to reviewer 1:

Dear the reviewer 1:

We highly appreciate your works and valuable suggestions on the revisions of our manuscript. According to the comments, we have carried out further calculations and revised the manuscript carefully, the corresponding replies are listed point-by-point along with the comments as follows:

Comment 1): In the manuscript, the outer edges of the monomer RRM12 domains were divided into four regions (i.e., shoulder-a, shoulder-b, head-1 and head-2 regions) to identify the types of different oligomer models (see figure 2b). Specifically, the shoulder- α region consisted of two α 2-helices in RRM1 and RRM2 domains, and the linking loop between RRM1 and RRM2 domains; The head-1 region consisted of α 1 and α 2 helices in RRM1 domain. According to the above definitions, the shoulder- α and head-1 regions were overlapped at α 2 helix. Why the authors chose this set of definition instead of dividing the RRM12 domains into separated regions?

Author reply:

Thank you for pointing out this problem. It is our negligence about the definition of the shoulder and head sides to make such overlapping in the previous manuscript. According to the comment and to make the definition more clear, we redefined the four outer edges as shoulder12-a, shoulder12-b, head-1 and head-2 regions, where “1” and “2” mean respectively the RRM1 and RRM2 domains, this definition would eliminate the previous overlap. The revised descriptions about the definition were shown on Page 10 Line 35 as follows:

“Based on the asymmetrical ellipsoid shape of the monomer RRM12 model (Figure 2b), four outer edges of each model were defined as “shoulder12- α ”, “shoulder12- β ”, and “head-1”, “head-2”, which will be applied to easily identify the types and names of the five dimer models. The shoulder12- α in each model represents the outer edge consisted of the α 2-helix in RRM2 domain, and the

linking loop (or the α -helix-linker) between RRM1 and RRM2 domains; the shoulder12- β represents the outer edge consisted of two β -sheet layers in RRM1 and RRM2 domains; then head-1 represents the outer edge constructed by the α 1, α 2 helixes of the RRM1 domain, and head-2 represents the outer edge constructed by the β -sheet layer tail of the RRM2 domain (Figure 2b).”

Comment 2): The authors are suggested to provide some secondary structure information of the equilibrated oligomers.

Author reply:

According to the comment, we added more secondary structure information of the equilibrated dimer and tetramer models on Page 20 Line 12 as follows:

“Moreover, the secondary structures were changed during the dimerization from the monomer RRM12 model to some dimer models. For example, it is found through comparing the average structures of the RRM1 and RRM2 domains in the monomer RRM12 with the dimer $S_{\beta}S_{\beta}/D$ model (see Figure 2b and Figure 3b) that the loose β -strand link between α 2 and β 5 in monomer forms the stable β 4-strand (from His166 to Ile168) and the β 5 length of RRM1 domain is increased from three residues (from Asp174 to Lys176) in monomer to five residues (from Arg171 to Cys176) in the $S_{\beta}S_{\beta}/D$ model (greened in Figure 3b). However, the length of α 2 helix with seven residues (from Asp237 to Leu243) in RRM2 domain of monomer RRM12 decreases to that with four residues (from Ala240 to 243Leu); the α -helix linking loop between the RRM1 and RRM2 domains in monomer RRM12 model changes to the loop linking conformation in the same dimer $S_{\beta}S_{\beta}/D$ model.”

Comment 3): The manuscript suggested that the aggregation coming from RRM1 domains was dominant and that from the RRM2 domain was second choice (page 15). Can the authors give some explanations based on the sequences of RRM1 and RRM2

domain?

Author reply:

Thank the reviewer for raising this good question. According to the comment, we added some explanations for the aggregation coming from RRM1 and RRM2 domains based on the sequences of RRM1 and RRM2 domains on Page 16 Line 39 as follows:

“For this reason, it is found that the residues of Leu109 of β 1, Asp165 of β 4, Arg171, Asp174, Lys176, Leu177 of β 5 on the β -sheet layers in the RRM1 domain might be the main contribution to the oligomer aggregation; accordingly, these residues are absent in the RRM2 domain for the $S_{\alpha}S_{\beta}/D$ model. This result is consistent with the experimental fact that the β -sheet layers of the RRM1 domain is an important motif with the highly conserved sequence for RNA binding ability and the pathological of TDP-43 aggregation [57].”

Comment 4): Models of monomers, dimers, tetramers and larger oligomers of RRM12 domains in TDP-43 have been obtained in this manuscript. So, is there a critical oligomer size above which the oligomers are stable, like the aggregation process of human islet amyloid polypeptide (e.g., PMID: 30502402)?

Author reply:

Thank the reviewer for raising this question that makes our study more meaningful. We have carefully read the reference and explored the aggregation properties of our studied models. It is found that the aggregation stability mainly comes from the interaction between the outer edges of the RRM12 domains of TDP-43 proteins due to the slight changes of secondary structures among each domain in the aggregation process. According to the comment, we added some corresponding expressions about the aggregation properties as follows:

On Page 5 Line 35:

“Feng Ding and co-workers theoretically studied the oligomerization of full length hIAPP through multiple molecular systems of increasing number of peptides and suggested that oligomers larger than a trimer allowed the formation of more stable β -sheets of the human islet amyloid polypeptide [44].”

On Page 18 Line 25:

“This result indicates that there is the concerted interaction affinity between the shoulder-to-shoulder and head-to-head interfaces. However, since it is the head-to-head interaction that contributes to the growth of the polymerization, it is expected this concerted interaction affinity will reach a constant as the number of the head-to-head interfaces produced by the aggregation process becomes large enough.”

Appendix B

Reply letter to reviewer 2:

Dear the reviewer 2:

We highly appreciate your works and valuable suggestions on the revisions of our manuscript. According to the comments, we have carried out further calculations and revised the manuscript carefully, the corresponding replies are listed point-by-point along with the comments as follows:

Comment (i): I would not consider this study as a molecular dynamics study. Instead, it is essentially docking based study using ZDOCK software to model the dimer, tetramer, etc.

Author reply:

Thanks for mentioning this points. It is true that the paper is intended to suggest a model on which to explain the oligomerization of TDP-43. After considering, we think it is better to renew the paper title as “**Insights into the aggregation mechanism of RRM domains in TDP-43: A theoretical exploration**”

Comment (ii): The authors superpose isolate RRM1 and RRM2 to RNA-bound dimeric RRM12 structure (pdb: 4BS2) and then removed RNA. They have constructed the intermediate loops (Lys102-ASN267 and ASN179 to Val193). For this, they mentioned using NMR experimental data but didn't refer to which NMR data. After the loop reconstruction, they mutated 200Gly to 200Glu.

Author reply:

According to the comment, we corrected description about the construction of the monomer RRM12 model and gave the NMR experimental data with the PDB code number, and also added the reason about the mutation of Gly200 residue recovered by Glu200 residue in the RRM2 domain, i.e. we recovered the first sequence of RRM2 domain and gave the corresponding reference on Page 6 Line 39 as follows:

“Namely, first, the basis structure of RRM12 model was taken from the NMR structure of the RNA binding RRM domains (PDB code: 4BS2) with the direct superposing of isolate RRM1 and RRM2 on the corresponding positions; second, the corrected coordinates of RRM1 and RRM2 domains captured from that of the NMR structure of isolate RRM1 and RRM2 (PDB code: 2CQG and 1WF0) were merged into the corresponding positions of NMR structure of the RNA binding RRM domains (PDB code: 4BS2); finally, besides removing the bound RNA structure, the full residue coordinates from Lys102 to Asn267 of the RRM1 and RRM2 domains and the linking loop of Asn179 to Val193 between the two domains in the RRM12 model were constructed by the NMR experimental data (PDB code: 4BS2) with the deletion of the extra residues in the RRM1 domain and with the mutated Gly200 residue recovered by the original Glu200 residue in the RRM2 domain for recovering the primary sequence of this domain [45].”

Comment (iii): They adopted such extensive structural reconstruction to build the dimeric model, to begin with. Under such new changes, the system certainly demands a long-time relaxation at an atomistic level. The authors claim the equilibrium reaches immediately after 40ns and their equilibrium statistics range is only from their 40-50ns (10 ns span) for the dimer. This small timescale is unacceptable considering the timescale required for minimum structural rearrangement (of the order of μ s) of such a complex dimeric system under the new reconstruction.

Author reply:

Thank the reviewer for raising this question that makes our study more meaningful. We performed the the MD simulation runs of 1 μ s for all dimer models. The corresponding properties for these models were extracted from the trajectory analysis between 800 ns and 1000 ns of simulation time. The corresponding data and figures have been updated through the whole manuscript. In general, it is found that the conclusions summarized from the revised computation are in consistent with those from the original one, in spite that the individual data are a little different from each other due to the difference of simulation times.

Comment (iv): For such a large system they have taken only 8 Å solvation box. In this confined boundary condition, I am afraid of whether any rotation/translational movements are at all feasible to facilitate any conformational rearrangement to relax the structure.

Author reply:

It is quite possible that we didn't clearly describe the water solvation model in the manuscript. In fact, the model system was enclosed by a rectangular water shell with the depth of the shell to be 8 Å in all the six directions of this model.

According to the suggestion of the reviewer, we extended the depth of the shell to be 14 Å. The descriptions of the model were also changed on Page 7 Line 21 as follows:

“For MD simulation, each model system was explicitly solvated by using the transferable intermolecular potential 3P (TIP3P) water inside a rectangular box large enough to ensure the solvent shell with the depth of 14 Å in all the six directions of the model.”

”

Comment (v): On page 8, after getting the docked structures, the authors calculate binding free energy using MM-PBSA where they mentioned that this free energy only includes enthalpy term as entropy requires much more computational resources. They also mentioned that the entropy term could be neglected when the receptor and ligand have similar conformations.

This cannot be true as the structure has a number of flexible linkers. Conformational changes are inevitable for such a hybrid structure where disordered segments would substantially be perturbed after binding to another monomer. They have deliberately ignored conformational dynamics during any sort of oligomer aggregation process.

I do not recommend this for publication unless the models are appropriately simulated using a reasonable box size for a longer time scale (at least 1μs because of the system's size) for relaxation and an appropriate free energy calculation at least for the dimer. Without appropriate analyses of the involved conformational changes during such oligomer aggregation process, such study will not provide any correct insight.

Author reply:

We appreciate the reviewer for raising the question about the importance of the entropy. Although the enthalpy can generally be employed for discussion instead of entropy in our existing experience, the recalculated results indicate that the entropy is important for the present studied system.

According to the suggestions of the reviewer, we managed to recalculate the MM-PBSA with the entropy terms for all dimer, tetramer and hexamer models based on the simulations with time of 1 μ s and a water box with the side length as large as 14 Å. As mentioned by the reviewer, the entropy calculation is important for these investigated aggregation models due to the fact that the structure has a number of flexible linkers. The Gibbs free energy for all the conformations are accordingly calculated from the enthalpies and entropies. The recalculated data have been updated for all models and discussions in the manuscript. Fortunately, the tendency of stabilities of the models obtained from the recalculated binding free energies with the entropy terms are in consistent with those of the original ones.

It is our carelessness that we didn't summarize the conformational changes during the oligomer aggregation process, although they were discussed separately in the manuscript. We summarized the discussions on the conformational changes of the oligomer aggregation process on Page 22 Line 19 as follows:

“Summarily, the studied oligomer aggregation process presents two types of combination of out edges of the two domains of TDP-43 protein, i.e. one is the shoulder-to-shoulder type which occurs at the interface between the two β -sheet layers or between one β -sheet layer and the α -helix linking loop in the different shoulder sides; the other is the head-to-head type which occurs at the interface between the two α 1- α 2 helix planes or between the two β -sheet layer tails in the different head sides. Consequently, the conformational changes mainly involve the tertiary structures, but the secondary structures were changed slightly in the dimer, tetramer and hexamer models. For example, the mass center distances of 25.8 Å, 25.8 Å and 25.4 Å between the RRM1 and RRM2 domains in the dimer $S_\alpha S_\beta/D$, tetramer $S_\alpha S_\beta 1/T$ and hexamer $S_\alpha S_\beta/H$ models, respectively, are longer than that of 23.9 Å in the monomer RRM12 model, which expanded the expected

oligomer aggregation edges and induces the looser conformation of oligomer in the shoulder-to-shoulder interface, favoring the aggregation process.”

Appendix C

Dear Editor:

We highly appreciate your working on our revised paper entitled “Insights into the aggregation mechanism of RRM domains in TDP-43: A theoretical exploration”, Manuscript ID: RSOS-210160. We also thank the referees for the good suggestions and the work on the paper. We included in this letter the replies to the reviewer’s and the editor’s comments point-by-point. If you have any question about them, please feel free to contact me.

Thank you very much for your consideration.

Sincerely,

Chaoqun Li, Ph.D.

College of Chemistry, Chemical Engineering and Material

Handan University

Hebei province, P.R.China

Email: lichaoqun@hdc.edu.cn

Reply letter to reviewer 1:

Dear the reviewer 1:

We highly appreciate your works and valuable suggestions on the revisions of our manuscript. According to the comments, we have carried out further calculations and revised the manuscript carefully, the corresponding replies are listed point-by-point along with the comments as follows:

Comment 1): In the manuscript, the outer edges of the monomer RRM12 domains were divided into four regions (i.e., shoulder-a, shoulder-b, head-1 and head-2 regions) to identify the types of different oligomer models (see figure 2b). Specifically, the shoulder- α region consisted of two α 2-helices in RRM1 and RRM2 domains, and the linking loop between RRM1 and RRM2 domains; The head-1 region consisted of α 1 and α 2 helices in RRM1 domain. According to the above definitions, the shoulder- α and head-1 regions were overlapped at α 2 helix. Why the authors chose this set of definition instead of dividing the RRM12 domains into separated regions?

Author reply:

Thank you for pointing out this problem. It is our negligence about the definition of the shoulder and head sides to make such overlapping in the previous manuscript. According to the comment and to make the definition more clear, we redefined the four outer edges as shoulder12-a, shoulder12-b, head-1 and head-2 regions, where “1” and “2” mean respectively the RRM1 and RRM2 domains, this definition would eliminate the previous overlap. The revised descriptions about the definition were shown on Page 10 Line 35 as follows:

“Based on the asymmetrical ellipsoid shape of the monomer RRM12 model (Figure 2b), four outer edges of each model were defined as “shoulder12- α ”, “shoulder12- β ”, and “head-1”, “head-2”, which will be applied to easily identify the types and names of the five dimer models. The shoulder12- α in each model represents the outer edge consisted of the α 2-helix in RRM2 domain, and the

linking loop (or the α -helix-linker) between RRM1 and RRM2 domains; the shoulder12- β represents the outer edge consisted of two β -sheet layers in RRM1 and RRM2 domains; then head-1 represents the outer edge constructed by the α 1, α 2 helixes of the RRM1 domain, and head-2 represents the outer edge constructed by the β -sheet layer tail of the RRM2 domain (Figure 2b).”

Comment 2): The authors are suggested to provide some secondary structure information of the equilibrated oligomers.

Author reply:

According to the comment, we added more secondary structure information of the equilibrated dimer and tetramer models on Page 20 Line 12 as follows:

“Moreover, the secondary structures were changed during the dimerization from the monomer RRM12 model to some dimer models. For example, it is found through comparing the average structures of the RRM1 and RRM2 domains in the monomer RRM12 with the dimer $S_{\beta}S_{\beta}/D$ model (see Figure 2b and Figure 3b) that the loose β -strand link between α 2 and β 5 in monomer forms the stable β 4-strand (from His166 to Ile168) and the β 5 length of RRM1 domain is increased from three residues (from Asp174 to Lys176) in monomer to five residues (from Arg171 to Cys176) in the $S_{\beta}S_{\beta}/D$ model (greened in Figure 3b). However, the length of α 2 helix with seven residues (from Asp237 to Leu243) in RRM2 domain of monomer RRM12 decreases to that with four residues (from Ala240 to 243Leu); the α -helix linking loop between the RRM1 and RRM2 domains in monomer RRM12 model changes to the loop linking conformation in the same dimer $S_{\beta}S_{\beta}/D$ model.”

Comment 3): The manuscript suggested that the aggregation coming from RRM1 domains was dominant and that from the RRM2 domain was second choice (page 15). Can the authors give some explanations based on the sequences of RRM1 and RRM2

domain?

Author reply:

Thank the reviewer for raising this good question. According to the comment, we added some explanations for the aggregation coming from RRM1 and RRM2 domains based on the sequences of RRM1 and RRM2 domains on Page 16 Line 39 as follows:

“For this reason, it is found that the residues of Leu109 of β 1, Asp165 of β 4, Arg171, Asp174, Lys176, Leu177 of β 5 on the β -sheet layers in the RRM1 domain might be the main contribution to the oligomer aggregation; accordingly, these residues are absent in the RRM2 domain for the $S_{\alpha}S_{\beta}/D$ model. This result is consistent with the experimental fact that the β -sheet layers of the RRM1 domain is an important motif with the highly conserved sequence for RNA binding ability and the pathological of TDP-43 aggregation [57].”

Comment 4): Models of monomers, dimers, tetramers and larger oligomers of RRM12 domains in TDP-43 have been obtained in this manuscript. So, is there a critical oligomer size above which the oligomers are stable, like the aggregation process of human islet amyloid polypeptide (e.g., PMID: 30502402)?

Author reply:

Thank the reviewer for raising this question that makes our study more meaningful. We have carefully read the reference and explored the aggregation properties of our studied models. It is found that the aggregation stability mainly comes from the interaction between the outer edges of the RRM12 domains of TDP-43 proteins due to the slight changes of secondary structures among each domain in the aggregation process. According to the comment, we added some corresponding expressions about the aggregation properties as follows:

On Page 5 Line 35:

“Feng Ding and co-workers theoretically studied the oligomerization of full length hIAPP through multiple molecular systems of increasing number of peptides and suggested that oligomers larger than a trimer allowed the formation of more stable β -sheets of the human islet amyloid polypeptide [44].”

On Page 18 Line 25:

“This result indicates that there is the concerted interaction affinity between the shoulder-to-shoulder and head-to-head interfaces. However, since it is the head-to-head interaction that contributes to the growth of the polymerization, it is expected this concerted interaction affinity will reach a constant as the number of the head-to-head interfaces produced by the aggregation process becomes large enough.”

Reply letter to reviewer 2:

Dear the reviewer 2:

We highly appreciate your works and valuable suggestions on the revisions of our manuscript. According to the comments, we have carried out further calculations and revised the manuscript carefully, the corresponding replies are listed point-by-point along with the comments as follows:

Comment (i): I would not consider this study as a molecular dynamics study. Instead, it is essentially docking based study using ZDOCK software to model the dimer, tetramer, etc.

Author reply:

Thanks for mentioning this points. It is true that the paper is intended to suggest a model on which to explain the oligomerization of TDP-43. After considering, we think it is better to renew the paper title as “**Insights into the aggregation mechanism of RRM domains in TDP-43: A theoretical exploration**”

Comment (ii): The authors superpose isolate RRM1 and RRM2 to RNA-bound dimeric RRM12 structure (pdb: 4BS2) and then removed RNA. They have constructed the intermediate loops (Lys102-ASN267 and ASN179 to Val193). For this, they mentioned using NMR experimental data but didn't refer to which NMR data. After the loop reconstruction, they mutated 200Gly to 200Glu.

Author reply:

According to the comment, we corrected description about the construction of the monomer RRM12 model and gave the NMR experimental data with the PDB code number, and also added the reason about the mutation of Gly200 residue recovered by Glu200 residue in the RRM2 domain, i.e. we recovered the first sequence of RRM2 domain and gave the corresponding reference on Page 6 Line 39 as follows:

“Namely, first, the basis structure of RRM12 model was taken from the NMR structure of the RNA binding RRM domains (PDB code: 4BS2) with the direct superposing of isolate RRM1 and RRM2 on the corresponding positions; second, the corrected coordinates of RRM1 and RRM2 domains captured from that of the NMR structure of isolate RRM1 and RRM2 (PDB code: 2CQG and 1WF0) were merged into the corresponding positions of NMR structure of the RNA binding RRM domains (PDB code: 4BS2); finally, besides removing the bound RNA structure, the full residue coordinates from Lys102 to Asn267 of the RRM1 and RRM2 domains and the linking loop of Asn179 to Val193 between the two domains in the RRM12 model were constructed by the NMR experimental data (PDB code: 4BS2) with the deletion of the extra residues in the RRM1 domain and with the mutated Gly200 residue recovered by the original Glu200 residue in the RRM2 domain for recovering the primary sequence of this domain [45].”

Comment (iii): They adopted such extensive structural reconstruction to build the dimeric model, to begin with. Under such new changes, the system certainly demands a long-time relaxation at an atomistic level. The authors claim the equilibrium reaches immediately after 40ns and their equilibrium statistics range is only from their 40-50ns (10 ns span) for the dimer. This small timescale is unacceptable considering the timescale required for minimum structural rearrangement (of the order of μ s) of such a complex dimeric system under the new reconstruction.

Author reply:

Thank the reviewer for raising this question that makes our study more meaningful. We performed the the MD simulation runs of 1 μ s for all dimer models. The corresponding properties for these models were extracted from the trajectory analysis between 800 ns and 1000 ns of simulation time. The corresponding data and figures have been updated through the whole manuscript. In general, it is found that the conclusions summarized from the revised computation are in consistent with those from the original one, in spite that the individual data are a little different from each other due to the difference of simulation times.

Comment (iv): For such a large system they have taken only 8 Å solvation box. In this confined boundary condition, I am afraid of whether any rotation/translational movements are at all feasible to facilitate any conformational rearrangement to relax the structure.

Author reply:

It is quite possible that we didn't clearly describe the water solvation model in the manuscript. In fact, the model system was enclosed by a rectangular water shell with the depth of the shell to be 8 Å in all the six directions of this model.

According to the suggestion of the reviewer, we extended the depth of the shell to be 14 Å. The descriptions of the model were also changed on Page 7 Line 21 as follows:

“For MD simulation, each model system was explicitly solvated by using the transferable intermolecular potential 3P (TIP3P) water inside a rectangular box large enough to ensure the solvent shell with the depth of 14 Å in all the six directions of the model.”

”

Comment (v): On page 8, after getting the docked structures, the authors calculate binding free energy using MM-PBSA where they mentioned that this free energy only includes enthalpy term as entropy requires much more computational resources. They also mentioned that the entropy term could be neglected when the receptor and ligand have similar conformations.

This cannot be true as the structure has a number of flexible linkers. Conformational changes are inevitable for such a hybrid structure where disordered segments would substantially be perturbed after binding to another monomer. They have deliberately ignored conformational dynamics during any sort of oligomer aggregation process.

I do not recommend this for publication unless the models are appropriately simulated using a reasonable box size for a longer time scale (at least 1μs because of the system's size) for relaxation and an appropriate free energy calculation at least for the dimer. Without appropriate analyses of the involved conformational changes during such oligomer aggregation process, such study will not provide any correct insight.

Author reply:

We appreciate the reviewer for raising the question about the importance of the entropy. Although the enthalpy can generally be employed for discussion instead of entropy in our existing experience, the recalculated results indicate that the entropy is important for the present studied system.

According to the suggestions of the reviewer, we managed to recalculate the MM-PBSA with the entropy terms for all dimer, tetramer and hexamer models based on the simulations with time of 1 μ s and a water box with the side length as large as 14 Å. As mentioned by the reviewer, the entropy calculation is important for these investigated aggregation models due to the fact that the structure has a number of flexible linkers. The Gibbs free energy for all the conformations are accordingly calculated from the enthalpies and entropies. The recalculated data have been updated for all models and discussions in the manuscript. Fortunately, the tendency of stabilities of the models obtained from the recalculated binding free energies with the entropy terms are in consistent with those of the original ones.

It is our carelessness that we didn't summarize the conformational changes during the oligomer aggregation process, although they were discussed separately in the manuscript. We summarized the discussions on the conformational changes of the oligomer aggregation process on Page 22 Line 19 as follows:

“Summarily, the studied oligomer aggregation process presents two types of combination of out edges of the two domains of TDP-43 protein, i.e. one is the shoulder-to-shoulder type which occurs at the interface between the two β -sheet layers or between one β -sheet layer and the α -helix linking loop in the different shoulder sides; the other is the head-to-head type which occurs at the interface between the two α 1- α 2 helix planes or between the two β -sheet layer tails in the different head sides. Consequently, the conformational changes mainly involve the tertiary structures, but the secondary structures were changed slightly in the dimer, tetramer and hexamer models. For example, the mass center distances of 25.8 Å, 25.8 Å and 25.4 Å between the RRM1 and RRM2 domains in the dimer $S_\alpha S_\beta/D$, tetramer $S_\alpha S_\beta 1/T$ and hexamer $S_\alpha S_\beta/H$ models, respectively, are longer than that of 23.9 Å in the monomer RRM12 model, which expanded the expected

oligomer aggregation edges and induces the looser conformation of oligomer in the shoulder-to-shoulder interface, favoring the aggregation process.”

Appendix D

Dear Editor:

We heartily appreciate your working on our revised paper entitled “Insights into the aggregation mechanism of RRM domains in TDP-43: A theoretical exploration”, Manuscript ID: RSOS-210160.R1. We also thank the referees for the helpful suggestions and the work on the paper. We included in this letter the replies to the reviewer’s comments point-by-point. If you have any question about them, please feel free to contact me.

Thank you very much for your consideration.

Sincerely,

Chaoqun Li, Ph.D.

College of Chemistry, Chemical Engineering and Material

Handan University

Hebei province, P.R.China

Email: lichaoqun@hdc.edu.cn

Reply to the review comments:

Comment (i): In Figure 2A RMSD values of heavy atoms need to be updated from 50 ns to 1 μ s.

Author reply:

According to the comment, the RMSD values of heavy atoms in Figure 2A have been updated to 1 μ s.

Comment (ii): Figure 7b, why dynamic cross has been calculated from the first 10 ns? Why not from the equilibrium trajectory extracted from 1 μ s?

Author reply:

Thank the reviewer for pointing out this problem. It is quite possible that we didn't clearly describe it in the manuscript. Because the dynamic cross calculation is employed to describe the motion correlations of the allosteric communication, the simulation in the early times specifically show the allosteric characteristics in the aggregation process from the dimer $S_{\alpha}S_{\beta}/D$ model to tetramer $S_{\alpha}S_{\beta}1/T$ model. We modified the corresponding description on Page 17 Line 8 as follows:

“To address further the stability of affinity at the shoulder-to-shoulder interface in the tetramer $S_{\alpha}S_{\beta}1/T$ model, the motion correlations of the allosteric communication for the backbone atoms during the first 10 ns simulation time in the aggregation process from the dimer $S_{\alpha}S_{\beta}/D$ model to tetramer $S_{\alpha}S_{\beta}1/T$ model have been calculated, and the corresponding cross correlation map was shown in Figure 7b.”

Comment (iii): Figure 10 (a): Time-dependences of distances should be plotted for the whole 1 μ s.

Author reply:

According to the comment, the time-dependences of distances in Figure 10 has been plotted for the whole 1 μ s for all dimer models. Those for two tetramer models have been plotted for the whole 50 ns simulations because of the facts: (1) it is really difficult for us to perform the simulation time for 1 μ s for the tetramer models due to the huge computational demands; and (2) according to the results for the simulations of the dimer models, the data from 50 ns simulations could represent those from 1 μ s simulations; this explanation has also been raised in the first revised paper according to the reviewer 2's comment. The corrected figures and data have been changed in Figure 10a, b and in this revised manuscript.

Comment (iv): In the abstract this sentence needs be rewritten in meaningful way: "The parallel β -sheet layers between the RRM1 domains in these oligomer models, which provide the potential binding sites in the aggregation process, are formed energetically favorable."

Author reply:

Thank the reviewer for pointing out this meaningless sentence. We have corrected this sentence in meaningful way in the abstract section on Page 3 Line 18 as follows:

"The parallel β -sheet layers between the RRM1 domains provide most of the binding sites in these oligomer models, and thus play an important role in the aggregation process."